# A Simple "Motivation" Can Enhance Reinforcement Finetuning of Large Reasoning Models

**Junjie Zhang[1]**[*], **Guozheng Ma[1]**, **Shunyu Liu[1]**, **Haoyu Wang[1]**, **Jiaxing Huang[1]**,
**Ting-En Lin[2]**, **Fei Huang[2]**, **Yongbin Li[2]**[†], **Dacheng Tao[1]**[†]

[1]Generative AI Lab, College of Computing and Data Science, Nanyang Technological University, Singapore 639798
[2]Tongyi Lab, Alibaba Group

## Abstract

Reinforcement Learning with Verifiable Rewards (RLVR) has emerged as a powerful learn-to-reason paradigm for large reasoning models to tackle complex tasks. However, the current RLVR paradigm is still not efficient enough, as it works in a trial-and-error manner. To perform better, the model needs to explore the reward space by numerously generating responses and learn from fragmented reward signals, blind to the overall reward patterns. Fortunately, verifiable rewards make the natural language description of the reward function possible, and meanwhile, LLMs have demonstrated strong in-context learning ability. This motivates us to explore if large reasoning models can benefit from a **motivation** of the task, *i.e.*, awareness of the reward function, during the reinforcement finetuning process, as we humans sometimes do when learning. In this paper, we introduce *Motivation-enhanced Reinforcement Finetuning* (**MeRF**), an intuitive yet effective method enhancing reinforcement finetuning of LLMs by involving *"telling LLMs rules of the game"*. Specifically, **MeRF** directly injects the reward specification into the prompt, which serves as an in-context motivation for the model to be aware of the optimization objective. This simple modification leverages the in-context learning ability of LLMs, aligning generation with optimization, thereby incentivizing the model to generate desired outputs from both inner motivation and external reward. Empirical evaluations demonstrate that **MeRF** achieves substantial performance gains over the RLVR baseline. Moreover, ablation studies show that MeRF performs better with greater consistency between the in-context motivation and the external reward function, while the model also demonstrates an ability to adapt to misleading motivations through reinforcement finetuning.

## 1 Introduction

Large Language Models (LLMs) have demonstrated remarkable capabilities across a wide range of natural language understanding and generation tasks, such as instruction following (Ouyang et al., 2022; Zhou et al., 2023; Taori et al., 2023; Zhu et al., 2024; Tu et al., 2025b; Zhu et al., 2025), code generation (Chen et al., 2021; Nijkamp et al., 2022; Zhuo et al., 2024), and medical diagnosis (Singhal et al., 2025; Zhang et al., 2023; Wang et al., 2023). To further improve the reasoning capabilities of LLMs, Reinforcement Learning with Verifiable Rewards (RLVR) (Lambert et al., 2024; Team et al., 2025) has emerged as a promising alternative to conventional supervised fine-tuning approaches (Radford et al., 2018; Brown et al., 2020; Zhang et al., 2025a), as demonstrated by DeepSeek-R1 (Guo et al., 2025) and OpenAI-o1 (Jaech et al., 2024). RLVR treats reasoning as a sequential decision-making process and optimizes models using objective reward signals that can be automatically verified with explicit rules, such as matching mathematical answers to ground truth (Lambert et al., 2024; He et al., 2024), or checking code correctness through unit tests (Le et al., 2022; Liu et al., 2023). By optimizing models toward meeting the clearly defined success

---

[*]Email: junjie.zhang@ntu.edu.sg
[†]Corresponding authors: shuide.lyb@alibaba-inc.com, dacheng.tao@ntu.edu.sg

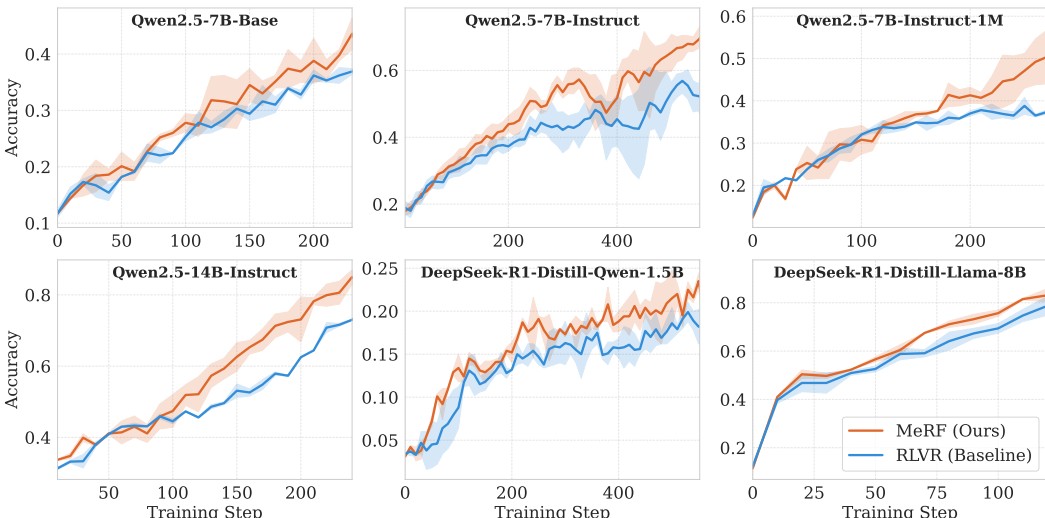

Figure 1: Validation Accuracy of **MeRF** and RLVR baseline on K&K Logic Puzzles in the training process. By simply *"telling LLMs rules of the game"* with in-context motivation during RL training, **MeRF** significantly outperforms the RLVR baseline with faster improvements, demonstrating the effectiveness of leveraging in-context motivation for more efficient RL training of LLMs.

criteria, RLVR enables LLMs to iteratively refine their reasoning capabilities, achieving significant performance improvements on complex tasks.

Following and extending the success of the conventional RL paradigm in gaming, Go, and robotics to the field of language models, RLVR relies on external reward signals to guide the training process of LLMs (Zeng et al., 2025; Yu et al., 2025; Hu et al., 2025; Chen et al., 2025; Zhang et al., 2025b; Fang et al., 2025). However, in the same way as conventional reinforcement learning, RLVR is not efficient enough, as it works in a trial-and-error manner, where the model can suffer from the sparse reward space and needs to learn the patterns of the task and reward function by repeatedly collecting and comparing its outputs and corresponding rewards. It requires a larger amount of training data and computational resources for LLMs to learn and generate enough responses to reach the expected behavior, and then get positive feedback. As we continue expecting LLMs to solve more and more complex tasks, the reward function can be more sophisticated and the expected behavior can be harder to reach, making it even more important to improve the efficiency of the current RLVR paradigm.

In RLVR, the model receives feedback on the correctness of its outputs, but lacks explicit awareness of the optimization objectives during training. As the verifiable reward function (with explicit rules) has demonstrated the desired behavior of the model, which can be described in natural language, and the in-context learning ability enables LLMs to learn from the given context, an intuitive question is **why not tell the LLMs what is the expected behavior, or how is their output get evaluated, during the training?** This is similar to how humans learn: when we have a task, we often benefit from understanding the rules and objectives before we start working on it. This understanding helps us to align our efforts with the desired outcomes, resulting in more efficient and effective learning.

In this paper, we introduce *Motivation-enhanced Reinforcement Finetuning* (**MeRF**), a simple yet powerful method that injects the reward specification directly into the prompt, serving as an in-context motivation for the model to be aware of the optimization objective. Unlike current RLVR paradigm leaving the model blind to the optimization objective during generation, relying on the transcendent reward function to guide the training process, **MeRF** explicitly informs the model about the reward structure and what constitutes a good response with in-context motivation, incentivizing the model to generate desired outputs from both inner motivation and external reward, leading to more efficient reinforcement finetuning as shown in Figure 1.

Our core contributions are summarized as follows:

- We propose **MeRF**, a novel motivation-enhanced reinforcement finetuning method for LLMs, enabling the model to be aware of the optimization objective by in-context motivation, to achieve more efficient and effective reinforcement finetuning of large reasoning models.

- Extensive experiments on the reasoning benchmarks: K&K Logic Puzzles, AIME24&25, AMC23, MATH500, and CountDown, show that **MeRF** significantly outperforms the RLVR baseline, validating its effectiveness and efficiency in improving reasoning capabilities in complex tasks.

- We provide a comprehensive analysis on the effectiveness of **MeRF**, including the impact of the consistency between the in-context motivation and the actual reward function, offering insights into the aligned in-context learning with reinforcement finetuning for self-evolving LLMs.

## 2 RELATED WORK

**Reinforcement Learning for LLMs.** RL has become a powerful paradigm for fine-tuning LLMs, first demonstrated by reinforcement learning from human feedback (RLHF) to align models with human-preferred responses (Ouyang et al., 2022; Rafailov et al., 2023; Liu et al., 2025). More recently, inspired by the success of DeepSeek-R1 (Guo et al., 2025), Reinforcement Learning with Verified Reward (RLVR) (Lambert et al., 2024) has been proposed to directly enhance reasoning performance by rewarding verifiable success criteria instead of subjective human judgments, which is more similar to the traditional RL paradigm (Jiang et al., 2021; 2022). RLVR has shown promising results in improving the performance of LLMs on various reasoning tasks, including logical reasoning (Xie et al., 2025), coding (Le et al., 2022; Liu et al., 2023), and math problems (Zeng et al., 2025), with simple, designed reward functions. However, these approaches demonstrate the potential of the conventional RL paradigm for LLMs, which may not fully leverage the in-context learning abilities of LLMs, the key to the success of LLMs in previous works (Wei et al., 2022; Brown et al., 2020; Tu et al., 2025a). In this work, we propose a novel method **MeRF**, leveraging the in-context abilities of LLMs to enhance reinforcement finetuning for reasoning.

**In-context Learning.** In-context learning (ICL) refers to the ability of large language models to learn a task purely from examples and instructions provided in the prompt, without any gradient-based updates to the parameters (Brown et al., 2020). This ability has been shown to be effective in various tasks, including few-shot learning, zero-shot learning, and even one-shot learning (Wei et al., 2022). ICL is a key feature of LLMs, enabling them to generalize from a few examples and adapt to new tasks quickly. The success of ICL has led to the development of various prompting strategies, such as few-shot prompting, chain-of-thought prompting, and self-consistency prompting (Wang et al., 2022). In this work, we investigate the potential of ICL for reinforcement finetuning of LLMs, and propose a novel method **MeRF** to leverage the in-context abilities of LLMs to enhance reinforcement finetuning for reasoning, injecting the in-context motivation into the training process.

## 3 METHOD

In this section, we introduce the Motivation-enhanced Reinforcement Finetuning (**MeRF**) for more efficient reinforcement learning with verifiable rewards of large reasoning models, which enables LLMs to be aware of the objective of the task in the reward space by in-context motivation.

### 3.1 MOTIVATION-ENHANCED REINFORCEMENT FINETUNING

Reinforcement learning of LLMs performs in a similar way as the human learning process, gaining improvements by learning from the feedback from the environment. The feedback is critical for the learning process, for it determines the direction of the optimization, which is often in the form of rewards, reflecting how well the model performs on the task. However, it is not easy for the model to learn the reward patterns before exploring the reward space extensively, especially when the reward space is sparse and the expected behavior is hard to reach. It can be inefficient and computationally expensive for LLMs, and at the beginning, most of the responses generated are bad and non-rewarded. Models work in a blind way, i.e., randomly respond to the question, and somehow the precious fragments of sparse positive rewards help the model to learn the reward patterns and expected behavior. This RLVR paradigm is not efficient enough and sometimes even fails to work, when the current model can hardly generate any good responses to get better rewards.

The cause of the problem is: the model is optimized in an indirect way with a black-box manner: the model is unaware of the overall optimization objective of training during the generation, gaining reward signal information by fragments (reward samples by the exploration of the model), and through

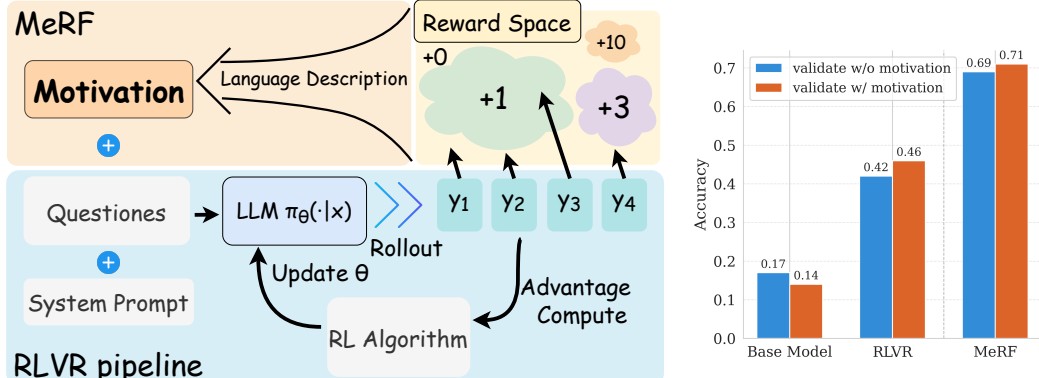

Figure 2: (Left) Illustration of the RLVR pipeline and the in-context motivation introduced by MeRF. Compared to the indirect way (reward samples generated and through parameter updates) to learn the reward patterns, MeRF enables the model to be aware of the overall reward space by in-context motivation. (Right) We validate the Base model, RLVR model and MeRF model on the K&K Logic Puzzle dataset in two settings: w/ motivation and w/o motivation in the prompt. Different from the base model, the RLVR model achieves a slightly better performance in validation w/ motivation than w/o motivation after the RLVR training, even while the motivation is not involved in the training process, indicating a connection between the **in-context motivation** validation and the RLVR training guided by the **reward function (as the motivation describes)**.

the parameter updates (policy gradient). The reward information can be local and one-sided when exploration fails to sufficiently cover the overall reward space (which is challenging), preventing the model from reaching the global optimum and leading instead to reward hacking toward local optima. As shown in Figure 2 (Left), in current RLVR pipeline, the model can't learn how to achieve the +10 reward if none of the generated responses can reach it, trapped in a paradoxical learning situation: **you need to learn something that you don't know how to do, or even don't know it exists**.

Fortunately, the problem can be alleviated in RLVR and LLMs. In the RLVR pipeline, the reward is verifiable with explicit rules, which means the reward function can be described by natural language, and meanwhile, LLMs have demonstrated strong in-context learning ability to learn from the given context. We propose to improve the information flow of reward signals in a more direct way, by **in-context motivation**, i.e., language description of the reward function, in the training process, to make the model aware of the optimization objective.

In Figure 2 (Right), we conduct an experiment to investigate how the in-context motivation affects the performance in the inference process and training process respectively. First, comparing the performance of the Base model with and without motivation in the inference time, Base model does not benefit from the motivation, indicating that the model can't learn the reward patterns and expected behavior by in-context motivation alone. Next, RLVR model (trained with the reward function described in the motivation), achieves an improvement when validated with in-context motivation. Moreover, when we involve the motivation during the training process (**MeRF**), there is a significant performance improvement over the RLVR baseline, demonstrating the effectiveness of MeRF in improving the efficiency and effectiveness of reinforcement finetuning of LLMs.

## 3.2 MOTIVATION WITH VERIFIABLE REWARD

Following the previous work (Xie et al., 2025) of RLVR, we utilize a verifiable reward function for the K&K Logic Puzzle dataset and demonstrate how the motivation is designed based on the reward function. The reward function contains two components: (i) Correctness Score and (ii) Format Score. It is implemented by rule-based verification and capable of being described in natural language, enabling the motivation to be injected into the training process. Here is the System Prompt and Motivation for K&K puzzle:

---

**System Prompt and Motivation for K&K Puzzle**

`<|im_start|>`system

You are a helpful assistant. The assistant first thinks about the reasoning process in the mind and then provides the user with the answer. The reasoning process and answer are enclosed within <think> </think> and<answer> </answer> tags, respectively, i.e., <think> reasoning process here </think><answer> answer here </answer>. Now the user asks you to solve a logical reasoning problem. After thinking, when you finally reach a conclusion, clearly state the identity of each character within <answer> </answer> tags. i.e., <answer> (1) Zoey is a knight (2) ... </answer>.

You will get evaluated following Evaluation Scoring Rules:
- Correctness Score:
    - If your final answer is correct, score 2
    - If your answer is understandable but wrong, score -1.5
    - If your answer is not parsable or incomplete, score -2
- Format Score:
    - If you follow the tag format exactly as above, score 1
    - Otherwise, score -1
You will get the final score as their sum. Example:
(1) The format follows the required structure: +1
(2) The final answer is correct: +2
(3) Total evaluation score: 3
Think carefully, follow the structure, and consider the evaluation rules.`<|im_end|>`

---

By injecting the motivation into the training process, we enable the model to be aware of the motivation of the task, which describes the reward function of the RLVR pipeline. The motivation provides a clear specification of what is expected and how to do it correctly, aligning the generation with the transcendent optimization objective. This approach leverages the in-context learning ability of LLMs to improve their reasoning capabilities in a more efficient and effective manner. More details about the complete motivation prompt design for the tasks are provided in Appendix A.3.

## 4 EXPERIMENT

In this section, we present the experimental results of our proposed method **MeRF**. We compare the performance of **MeRF** with the RLVR baseline to demonstrate the effectiveness of MeRF in improving the reasoning capabilities of LLMs in the reinforcement finetuning process.

### 4.1 EXPERIMENTAL SETUP

**Models and RL Algorithm.** We conducted the experiments with Qwen2.5 series (Yang et al., 2024) and DeepSeek-R1-Distill series (Guo et al., 2025), including: Qwen2.5-7B-Base, Qwen2.5-7B-Instruct, Qwen2.5-7B-Instruct-1M, Qwen2.5-14B-Instruct, DeepSeek-R1-Distill-Qwen-1.5B, DeepSeek-R1-Distill-Llama-8B. The models are across different model sizes, model families, and instruction-tuning stages, allowing us to comprehensively evaluate the effectiveness of **MeRF** in enhancing reinforcement finetuning of LLMs. We use GRPO as the RL algorithm for reinforcement finetuning. GRPO is an effective and efficient reinforcement learning algorithm for LLMs finetuning, which is suitable for our experiments to demonstrate the effectiveness of **MeRF** and RLVR baseline in the reinforcement finetuning process without demanding excessive computational resources.

**Dataset.** We evaluate **MeRF** on K&K (Logic Puzzles), MATH benchmarks: AIME24&25, AMC23, MATH500 (Lightman et al., 2023), and CountDown (Number Game) (Pan et al., 2025). The K&K dataset contains 7 different difficulty levels of logic puzzles, ranging from 2 people to 8 people in the task. We utilize the 3 to 7 people puzzles for training, the corresponding test set for in-domain evaluation, and the 2 and 8 people puzzles for out-of-distribution (OOD) evaluation. There are 900 samples for training in each difficulty level and 100 samples for evaluation. The total number of samples in the training set is 4500, and 700 samples for evaluation. For MATH benchmarks, we follow the previous work (Yu et al., 2025), using a subset of the DAPO-Math-17K dataset for training

and evaluating on AIME24&25, AMC23, and MATH500. We modify the original prompt of the training data for motivation design. For CountDown, we follow the previous work (Pan et al., 2025), using the same training and evaluation set. CountDown is a number game, where the model needs to use the given numbers and arithmetic operations to reach the target number. The given numbers can only be used once, and the number of given numbers is 3 or 4 in our experiments.

Table 1: Performance comparison across models on tasks with varying difficulty by number of people of K&K Puzzles. **MeRF** demonstrates a significant improvement over the RLVR baseline in both in-domain and OOD scenarios. **Notably**, all the results are validated without in-context motivation.

| Model | Difficulty by Number of People | | | | | | | | Avg. |
| --- | --- | --- | --- | --- | --- | --- | --- | --- | --- |
| | 3 | 4 | 5 | 6 | 7 | Avg. | 2 (OOD) | 8 (OOD) | |
| o3-mini-high | 0.98 | 0.97 | 0.95 | 0.94 | 0.89 | 0.95 | 0.99 | 0.83 | 0.94 |
| o1-2024-12-17 | 0.51 | 0.38 | 0.38 | 0.35 | 0.30 | 0.38 | 0.83 | 0.20 | 0.42 |
| Deepseek-R1 | 0.73 | 0.77 | 0.78 | 0.75 | 0.88 | 0.78 | 0.91 | 0.83 | 0.81 |
| GPT-4o | 0.57 | 0.49 | 0.32 | 0.23 | 0.21 | 0.36 | 0.68 | 0.11 | 0.37 |
| GPT-4o-mini | 0.42 | 0.34 | 0.17 | 0.09 | 0.10 | 0.22 | 0.63 | 0.01 | 0.25 |
| NuminaMath-7B-CoT | 0.13 | 0.12 | 0.05 | 0.01 | 0.00 | 0.06 | 0.28 | 0.00 | 0.08 |
| Deepseek-Math-7B | 0.21 | 0.08 | 0.06 | 0.02 | 0.00 | 0.07 | 0.35 | 0.00 | 0.10 |
| Qwen2.5-7B-Base | 0.34 | 0.16 | 0.09 | 0.00 | 0.00 | 0.12 | 0.41 | 0.00 | 0.14 |
| +RLVR (Baseline) | 0.48 | **0.53** | 0.30 | 0.21 | **0.16** | 0.34 | 0.59 | 0.17 | 0.35 |
| +MeRF (Ours) | **0.54** | **0.53** | **0.45** | **0.36** | **0.16** | **0.41** | **0.70** | **0.18** | **0.42** |
| Qwen2.5-7B-Instruct | 0.24 | 0.10 | 0.06 | 0.04 | 0.04 | 0.10 | 0.43 | 0.00 | 0.13 |
| +RLVR (Baseline) | 0.68 | 0.67 | 0.57 | 0.43 | 0.22 | 0.51 | 0.71 | 0.28 | 0.51 |
| +MeRF (Ours) | **0.78** | **0.73** | **0.68** | **0.62** | **0.42** | **0.65** | **0.76** | **0.39** | **0.63** |
| Qwen2.5-7B-Instruct-1M | 0.40 | 0.25 | 0.11 | 0.06 | 0.02 | 0.17 | 0.49 | 0.01 | 0.19 |
| +RLVR (Baseline) | 0.52 | 0.44 | 0.36 | 0.18 | 0.12 | 0.32 | 0.60 | 0.16 | 0.34 |
| +MeRF (Ours) | **0.68** | **0.62** | **0.48** | **0.42** | **0.20** | **0.48** | **0.76** | **0.25** | **0.49** |
| Qwen2.5-14B-Instruct | 0.46 | 0.31 | 0.20 | 0.09 | 0.11 | 0.23 | 0.63 | 0.06 | 0.27 |
| +RLVR (Baseline) | 0.90 | 0.82 | 0.78 | 0.70 | 0.55 | 0.75 | 0.90 | 0.42 | 0.72 |
| +MeRF (Ours) | **0.95** | **0.92** | **0.84** | **0.78** | **0.66** | **0.83** | **0.99** | **0.65** | **0.83** |
| DeepSeek-R1-Distill-Qwen-1.5B | 0.08 | 0.01 | 0.00 | 0.00 | 0.00 | 0.02 | 0.30 | 0.00 | 0.06 |
| +RLVR (Baseline) | 0.20 | 0.16 | **0.16** | **0.10** | **0.04** | 0.13 | 0.29 | 0.01 | 0.14 |
| +MeRF (Ours) | **0.25** | **0.20** | 0.12 | 0.08 | **0.04** | **0.14** | **0.43** | **0.04** | **0.17** |
| DeepSeek-R1-Distill-Llama-8B | 0.26 | 0.22 | 0.14 | 0.06 | 0.05 | 0.15 | 0.24 | 0.11 | 0.15 |
| +RLVR (Baseline) | 0.92 | 0.89 | 0.84 | 0.79 | 0.64 | 0.82 | 0.90 | 0.58 | 0.79 |
| +MeRF (Ours) | **0.95** | **0.94** | **0.92** | **0.87** | **0.71** | **0.88** | **0.99** | **0.64** | **0.86** |

## 4.2 MAIN RESULTS

To demonstrate the effectiveness of **MeRF** in the reinforcement finetuning process, we compare the performance of MeRF with the RLVR baseline on the K&K Logic Puzzle. The results in Figure 1 show that **MeRF** consistently achieves a significant improvement over the RLVR baseline in the validation accuracy during the training process, across different model sizes and model families, revealing the remarkable effectiveness of the motivation-enhanced reinforcement finetuning, with simply injecting the in-context motivation.

The results in Table 1 present the validation accuracy in the evaluation of different difficulty levels, comparing the performance of **MeRF** with the RLVR baseline, startpoint model, and other well-known models. **MeRF** achieves a significant improvement on the logic reasoning tasks from the startpoint Qwen2.5-7B-Instruct, with only hundreds of training steps, outperforming the RLVR baseline and even some commercial models in all difficulty levels including OOD scenarios. The results of the other baseline models suggest that K&K logic puzzles are challenging tasks for LLMs, and unseen in the training of most models, which further proves the fitness of K&K logic puzzles for analyzing the reasoning capabilities of RLVR models in our experiments, and the effectiveness of **MeRF** in the reinforcement finetuning process.

Table 2: Comparison of RLVR baseline and **MeRF** across math reasoning datasets. Each dataset occupies one row, with both methods displayed in vertical blocks (pass@1/2/4/8). The last row summarizes the average performance.

| Dataset | RLVR (Baseline) | | | | MeRF (Ours) | | | |
|---|---|---|---|---|---|---|---|---|
| | pass@1 | pass@2 | pass@4 | pass@8 | pass@1 | pass@2 | pass@4 | pass@8 |
| AIME24 | 16.7 | 20.0 | 20.0 | 26.7 | **20.0** | **20.0** | **26.7** | **30.0** |
| AIME25 | **6.7** | **16.7** | **16.7** | 20.0 | **6.7** | 10.0 | 16.7 | **26.7** |
| AMC23 | 47.5 | 57.5 | 70.0 | 72.5 | **55.0** | **67.5** | **72.5** | **77.5** |
| MATH500 | 62.6 | 72.8 | 77.0 | 82.6 | **65.4** | **74.0** | **81.8** | **85.6** |
| **Average** | 33.38 | 41.75 | 45.93 | 50.45 | **36.78**$^{+3.40}$ | **42.88**$^{+1.13}$ | **49.43**$^{+3.50}$ | **54.95**$^{+4.50}$ |

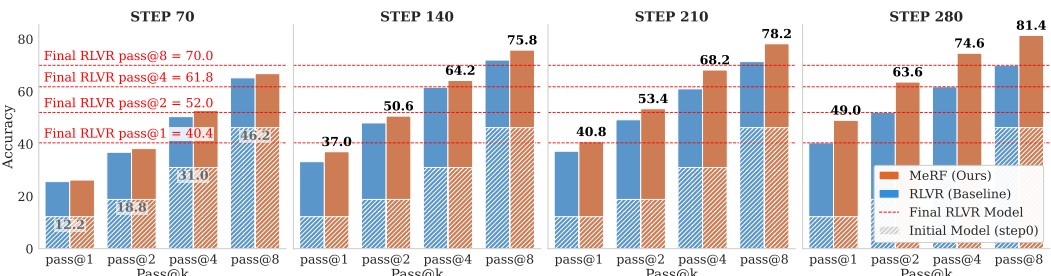

Figure 3: **Pass@$k$** performance of MeRF and RLVR baseline during the training process (from 0 to 280 steps) on K&K Logic Puzzle. We compare the pass@1, pass@2, pass@4, and pass@8 performance at each step, where MeRF consistently outperforms the RLVR baseline in all metrics. More importantly, MeRF demonstrates a significant training efficiency over RLVR baseline, for example, achieving better pass@4 and pass@8 performance at step 140 than the final RLVR model (at step 280), while RLVR's performance of pass@4 and pass@8 hardly improves after step 140.

Table 2 shows the performance comparison between **MeRF** and the RLVR baseline on MATH benchmarks, with Qwen2.5-7B-Base model as the startpoint. We report the pass@$k$ performance for $k \in \{1, 2, 4, 8\}$ on AIME24&25, AMC23, and MATH500 datasets, where MeRF achieves consistent improvements over the RLVR baseline in most metrics across all datasets, with an average gain of 3.40%, 1.13%, 3.50%, and 4.50% in pass@1, pass@2, pass@4, and pass@8, respectively. The results demonstrate the effectiveness of **MeRF** in enhancing the reasoning capabilities of LLMs in the reinforcement finetuning process on mathematical tasks.

## 5 ANALYSIS ON THE MECHANISM BEHIND MERF

In this section, we further analyze the effectiveness of MeRF by answering the following questions:

> Q1: Does the performance improvement of MeRF come from the in-context inference?

To answer this question, we conduct experiments to investigate how in-context motivation affects the performance in inference process and training process. As shown in Figure 2 (Right), we observe that for both RLVR and **MeRF** model, motivation validation leads to slightly better performance (4% and 2%) than non-motivation validation. However, compared to the performance improvement of **MeRF** over RLVR in both non-motivation validation and motivation validation (27% and 25%), the performance improvement of **MeRF** is mainly from the motivation-enhanced reinforcement finetuning process, demonstrating the effectiveness of the in-context motivation in the training process. The results in Figure 7 also show that the performance of both models does not benefit much from the in-context motivation validation, indicating that **the performance improvement of MeRF is not from the in-context inference.**

> **Q2:** If the performance improvement is not from in-context inference, how does the in-context motivation help to enhance the reinforcement finetuning process?

To further understand how the in-context motivation helps the reinforcement finetuning process, we analyze the pass@$k$ performance and entropy of the models during the training process. The results in Figure 3 show that **MeRF** consistently outperforms the RLVR baseline in all pass@$k$ metrics, demonstrating the effectiveness of MeRF in improving the reasoning capabilities of LLMs. More importantly, while RLVR's performance of pass@4 and pass@8 hardly improves after step 140, **MeRF** achieves better pass@4 and pass@8 performance at step 140 than the final RLVR Model (at step 280). The results in Figure 4 also show that **MeRF** outperforms the RLVR baseline consistently in both pass@8 and pass@1 metrics on the MATH500 dataset, while RLVR pass@8 performance hardly improves after step 80. Pass@$k$ metrics serve as an indicator of the model's ability

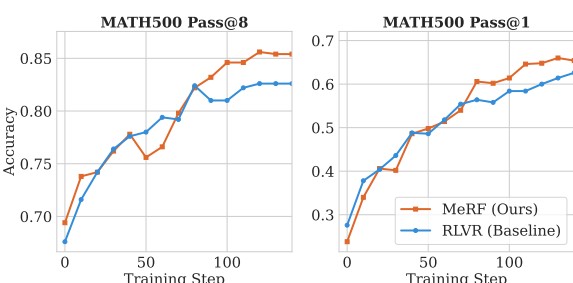

Figure 4: Comparison of **Pass@8 and Pass@1** performance of MeRF and RLVR baseline on MATH500 dataset during the training process. **MeRF** outperforms the RLVR baseline consistently in both pass@8 and pass@1 metrics, while RLVR pass@8 performance hardly improves after step 80, demonstrating the effectiveness of MeRF in improving the math reasoning capabilities of LLMs.

to explore diverse reasoning paths and reach a correct answer (Zeng et al., 2025; Shao et al., 2024). Better pass@$k$ performance and continuous pass@$k$ improvement during the training process suggest the model is more likely to reach a positive reward for optimization and performance improvement. As 8 is the rollout group size of GRPO in our experiments, the growing improvement of pass@8 performance of **MeRF** over the RLVR baseline indicates that **training process progressively amplifies the initial pass@8 improvement with better exploration ability, initially benefiting from the in-context motivation**, which also explains why the motivation validation only leads to slightly better performance than non-motivation validation but significant performance improvement of **MeRF** over RLVR baseline in Figure 2 (Right).

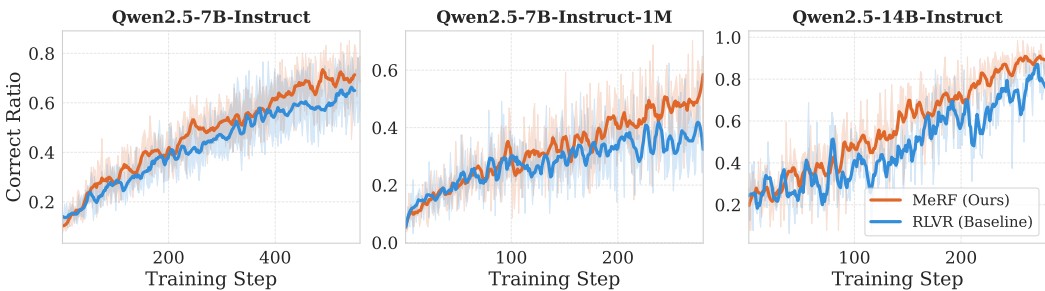

Figure 5: **Correct Ratio** of generated answers of the training set during the training process on K&K Logic Puzzle dataset. MeRF consistently outperforms the RLVR baseline, demonstrating the better exploration ability encouraged by the in-context motivation for the model to get the best reward during the training process.

As the result shown in Figure 6, **MeRF** maintains a higher entropy than the RLVR baseline. Specifically, we find that at the beginning of training, MeRF has lower entropy than RLVR, indicating that the structured exploration introduced by motivations helps the model to focus on more promising areas of the output space compared with naive exploration. As training progresses, RLVR entropy decreases more rapidly, indicating that the model is losing exploration ability and somehow collapsing (Cui et al., 2025) to suboptimal solutions (e.g., only get format reward) due to sparse reward signal and blind exploration. In contrast, MeRF maintains a relatively higher entropy throughout training, suggesting that the reward space information provided by motivations encourages the model

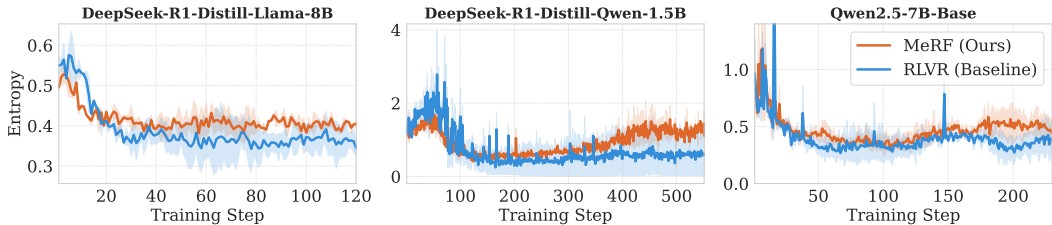

Figure 6: **Entropy** of models during the training process. MeRF maintains a higher entropy than the RLVR baseline, indicating that MeRF encourages more exploration by the in-context motivation during the training process, which contributes to its improved performance.

to explore more effectively for the best reward (correct answer, as shown in Figure 5) rather than collapsing to suboptimal solutions rapidly.

> **Takeaway:** The performance improvement of **MeRF** is not mainly from the in-context inference, but mainly from the training process. Training process progressively **amplifies** the initial pass@k improvement with better exploration ability, initially benefiting from the in-context motivation.

**Q3:** Does the training and validation gap (validation without motivation) affect the performance?

As **MeRF** includes the in-context motivation in the training process, while the **validation is conducted without motivation**, there exists a training and validation gap, which may affect the performance of MeRF. To investigate the impact of the training and validation gap on the performance, we conduct experiments to validate the models with and without motivation.

The results are shown in Figure 7, where we compare the performance of Qwen-2.5-7B-Instruct and DeepSeek-R1-Distill-Qwen-1.5B (Guo et al., 2025) trained with **MeRF** in the two validation settings. We find that both models achieve comparable performance between the two validation settings, indicating the negligible impact of the training and validation gap on the performance of these models. The results suggest that **the model is capable of generalizing to non-motivation validation when trained with in-context motivation**, which is essential for MeRF to be effective.

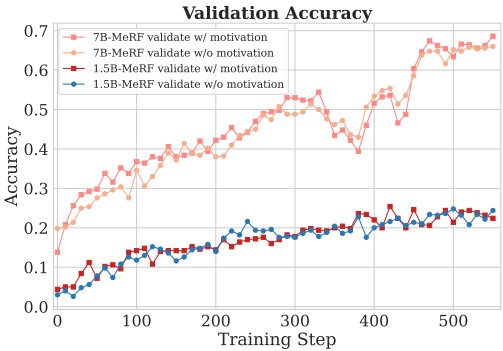

Figure 7: **Validation Accuray** w/ motivation and w/o motivation in the prompt for MeRF Models. The comparable performance between two validation settings of both MeRF Models, indicating the negligible impact of the training and validation gap in terms of the prompt, which does not necessarily lead to a drop in performance.

**Q4:** How do the different motivations (suboptimal, adverse) affect the performance?

We compare the performance of **MeRF** with different motivations, including the Motivation (Groud-Truth), which totally matches with the reward function of the optimization process, Motivation (Suboptimal), which is the suboptimal motivation only describing the correctness score, and Motivation (Adverse), which is the adverse motivation misleading the model to provide the wrong answer. The results are shown in Figure 8 (Left) and examples (segments) of the motivations are shown in Figure 8 (Right). The results demonstrate that the motivation with ground-truth reward function achieves the best performance, and the suboptimal motivation performs better than the RLVR baseline, with an additional correctness score description included in the motivation. Adverse motivation provides the full description of the reward function same to ground-truth motivation, while all the score is assigned to the opposite, which misleads the model to provide the wrong answer. The performance drop of the adverse motivation is mainly caused by the contradiction between the in-context motivation and the reward function of the optimization process. However, after several rounds of unstable learning

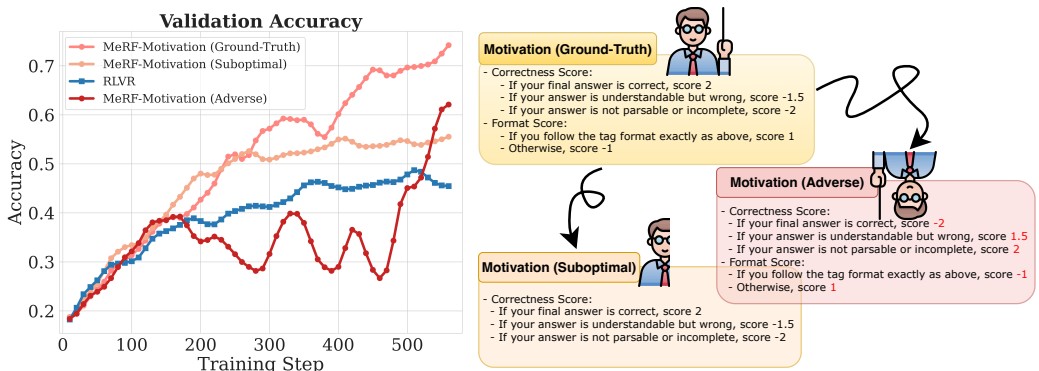

Figure 8: (Left) The performance of MeRF with **different motivations**. The motivation with ground-truth reward function achieves the best performance. Adverse motivation misleads the model to provide the wrong answer, while the model is capable of adapting to the adverse motivation in the process of reinforcement finetuning. (Right) Examples of different motivations. Suboptimal motivation only describes the correctness score, while adverse motivation provides the full description of the reward function with opposite scores.

dynamics (more results are provided in the Figure 10), the model adapted to discount the motivation signal, either treating the score as uninformative, or understanding the deliberately adverse motivation and opposite scores, while the full description of the reward function is still beneficial for the model to outperform the RLVR baseline and Motivation (Suboptimal). The results suggest that **MeRF benefits from a better consistency between the motivation and the underlying reward function**, resulting in better performance, while the model is capable of adapting to the adverse motivation in the process of reinforcement finetuning.

> **Takeaway:** The training and validation gap caused by in-context motivation has a negligible impact on the performance of models with strong generalization capabilities. **MeRF** benefits from a **better consistency** between the motivation and the underlying reward function, while the model is capable of adapting to the adverse motivation in the process of reinforcement finetuning.

## 6 CONCLUSION

In this paper, we propose **MeRF**, leveraging the in-context learning abilities of LLMs for more efficient reinforcement finetuning with in-context motivation. By injecting the in-context motivation into the training process, **MeRF** enables the model to be aware of the objective of the task, aligning the generation with the transcendent optimization objective, and therefore, leading to substantial performance improvement in reasoning benchmarks. To further understand the effectiveness of MeRF, we conduct comprehensive experiments and analysis, revealing the mechanism behind MeRF, demonstrating the powerful capability of LLMs in adapting to adverse motivation and the potential for more powerful large reasoning models with motivation-enhanced reinforcement finetuning.

**Limitations.** However, there are still some limitations in our work, presenting the potential for future research. (1) The motivation in MeRF is static in the training process. It is possible to explore the dynamic motivation during the training process in future work. (2) For models of weak generalization capability, how to efficiently implement RLVR and better leverage the in-context motivation for more efficient reinforcement finetuning is still an open question.

ACKNOWLEDGMENTS

This research is supported by the RIE2025 Industry Alignment Fund – Industry Collaboration Projects (IAF-ICP) (Award I2301E0026), administered by A*STAR, as well as supported by Alibaba Group and NTU Singapore through Alibaba-NTU Global e-Sustainability CorpLab (ANGEL).

ETHICS STATEMENTS

This work adheres to the ICLR Code of Ethics. Our research does not involve human subjects, personally identifiable information, or sensitive personal data. All datasets used are publicly available and commonly adopted in prior work; we followed their intended licensing and usage guidelines. We have carefully considered potential risks of misuse, including issues of fairness, bias, and safety, and we believe our contributions do not raise immediate ethical concerns.

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

# A  APPENDIX

## A.1  USE OF LLMS

We used Large Language Models (LLMs) as a general-purpose writing and editing tool to improve grammar, clarity, and readability of the manuscript. LLMs were not used for research ideation, experiment design, data analysis, or derivation of results. All technical contributions, experimental designs, and scientific claims were developed by the authors.

## A.2  PRELIMINARY

**Reinforcement Learning with Verifiable Rewards (RLVR)** (Lambert et al., 2024) is a reinforcement learning paradigm for training language models on tasks with verifiable outcomes, such as math problems or logic puzzles. The key idea is to use a reward function that can be automatically verified, allowing the model to learn from the ultimate correctness of its outputs. Recent works (Zeng et al., 2025; Yu et al., 2025) have shown that RLVR can significantly improve the reasoning capabilities of LLMs, with reasoning patterns emerging from the optimization for verifiable rewards.

**Group Relative Policy Optimization (GRPO)** (Shao et al., 2024) is a reinforcement learning algorithm designed to optimize policies by leveraging group-wise relative preference information. As a variant of Proximal Policy Optimization (PPO) (Schulman et al., 2017), GRPO foregoes the need for critic models and instead focuses on learning from relative comparisons of actions within groups, significantly enhancing training efficiency for reinforcement finetuning of LLMs. For each question $x$, GRPO samples a group of $G$ outputs $\{y_i\}_{i=1}^G$ from the policy $\pi_{\theta_{old}}(\cdot|x)$, and computes the advantage $A_i$ for each output $y_i$ based on the outcome reward $r_i$, where $\pi_{ref}$ is a reference model. The GRPO objective is defined as follows:

$$\mathcal{L}_{\text{GRPO}}(\theta) = \mathbb{E}_{x \sim \mathcal{D}, \{y_i\}_{i=1}^G \sim \pi_{\theta_{old}}(\cdot|x)} \left[ \frac{1}{G} \sum_{i=1}^G \min \left( \rho_i A_i, \text{clip}(\rho_i, 1 - \varepsilon, 1 + \varepsilon) A_i \right) \right]$$
$$- \beta \mathbb{D}_{\text{KL}} \left( \pi_\theta || \pi_{\text{ref}} \right), \tag{1}$$

where

$$\rho_i = \frac{\pi_\theta(y_i|x)}{\pi_{\theta_{old}}(y_i|x)} \tag{2}$$

is the importance ratio and the advantage is computed as:

$$A_i = \frac{r_i - \text{mean}(\{r_1, r_2, \cdots, r_G\})}{\text{std}(\{r_1, r_2, \cdots, r_G\})} \tag{3}$$

This normalizes the outcome rewards of the group of outputs and sets the advantage for all the tokens in the output $\{y_i\}_{i=1}^G$. This formulation enables GRPO to learn from relative preferences within each group without the need for a critic model, making it efficient and suitable for the implementation of RLVR to LLMs.

**The Knights and Knaves (K&K)** (Xie et al., 2024) logic puzzle dataset is a widely used (Xie et al., 2025) benchmark for reinforcement finetuning for LLMs reasoning, which provides a well-structured difficulty level of the logic tasks and allows accurate and easy reward verification for RLVR. The controllable difficulty levels are achieved by varying the number of people in the logic task, the more people in the logic puzzle requiring LLMs' more complex reasoning, and more steps to solve the task.

The K&K dataset contains 7 different difficulty levels of logic puzzles, with 2 people as the easiest level and 8 people as the most difficult level. Here is an example of the K&K dataset with 3 people:

---

**An Example of K&K Puzzle with 3 People**

**Problem:** A very special island is inhabited only by knights and knaves. Knights always tell the truth, and knaves always lie. You meet 3 inhabitants:
Penelope, David, and Zoey. Penelope noted, "David is a knight if and only if David is a knave". David told you that Zoey is a knave if and only if Zoey is a knight. According to Zoey, "If Penelope is a knave then David is a knave". So who is a knight and who is a knave?

**Solution:** (1) Penelope is a knave    (2) David is a knave    (3) Zoey is a knight

---

In this puzzle, the 3 inhabitants are either knights, who always tell the truth, or knaves, who always lie. The statements made by each inhabitant can be analyzed to determine their identities, leading to a unique and verifiable conclusion that Penelope and David are knaves, while Zoey is a knight. The K&K puzzles are challenging logic tasks systematically generated with logic templates, requiring multiple steps of reasoning and logical deduction to arrive at the correct answer. The complexity of the puzzles is precisely controllable by increasing the number of inhabitants. Moreover, the puzzles are unseen in the training of most models, combined with all the above, making it a suitable benchmark for evaluating the reasoning capabilities of RLVR LLMs.

**CountDown** (Pan et al., 2025) is a challenging numerical reasoning dataset that requires models to perform arithmetic operations and logical reasoning to arrive at the correct answer. The dataset consists of problems that involve a series of numbers and a target number, where the goal is to use the given numbers and basic arithmetic operations (addition, subtraction, multiplication, and division) to reach the target number. In our experiments, each problem provides a set of 3 or 4 numbers and a target number, and the model must determine a sequence of operations that will result in the target number. The problems in the CountDown dataset vary in difficulty, with some requiring simple calculations while others necessitate more complex reasoning and multiple steps to solve. The dataset is designed to test the model's ability to understand numerical relationships, perform calculations accurately, and apply logical reasoning to achieve the desired outcome. The CountDown dataset is widely used as a benchmark for evaluating the numerical reasoning capabilities of language models.

---

**An Example of CountDown dataset**

```
<|im_start|>system
```
You are a helpful assistant. The assistant first thinks about the reasoning process in the mind and then provides the user with the answer. The reasoning process and answer are enclosed within <think> </think> and <answer> </answer> tags, respectively, i.e., <think> reasoning process here </think><answer> answer here </answer>. Now the user asks you to solve a math reasoning problem. After thinking, when you finally reach a conclusion, clearly state the equation within <answer> </answer> tags. i.e., <answer> (1 + 2) / 3 </answer>. Now, the user will give you the math reasoning problem to solve. Think carefully, follow the structure.
```
<|im_end|>

<|im_start|>user
```
Using the numbers [2, 2, 2], create an equation that equals 8. You can use basic arithmetic operations (+, -, *, /) and each number can only be used once. `<|im_end|>`

```
<|im_start|>assistant
<think>
```

---

**Other Implementation Details**    We implement our method with VeRL (Sheng et al., 2024) and Logic-RL (Xie et al., 2025) codebase. We conduct the reinforcement finetuning with batch size of 16, a learning rate of 1e-6, 1 and 2 epochs of training. We use the rollout framework of Vllm (Kwon et al., 2023) for sampling the outputs of the model. The group size of GRPO is set to 8, the clip range is set to 0.2, and the KL penalty is set to 0.001.

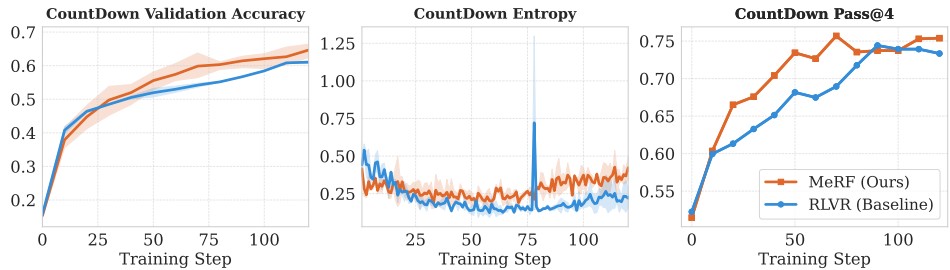

Figure 9: Validation Accuracy, Entropy, and Pass@4 on CountDown dataset. MeRF outperforms the RLVR baseline with better accuracy and pass@4 performance, while maintaining a higher entropy during the training process.

### A.3 Prompt Design of In-Context Motivation

Here is the System Prompt and Motivation for K&K Logic Puzzle, Math, and CountDown tasks used in our experiments.:

---

**System Prompt and Motivation for K&K Puzzle**

`<|im_start|>`system
You are a helpful assistant. The assistant first thinks about the reasoning process in the mind and then provides the user with the answer. The reasoning process and answer are enclosed within <think> </think> and<answer> </answer> tags, respectively, i.e., <think> reasoning process here </think><answer> answer here </answer>. Now the user asks you to solve a logical reasoning problem. After thinking, when you finally reach a conclusion, clearly state the identity of each character within <answer> </answer> tags. i.e., <answer> (1) Zoey is a knight (2) ... </answer>.
You will get evaluated following Evaluation Scoring Rules:
- Correctness Score:
   - If your final answer is correct, score 2
   - If your answer is understandable but wrong, score -1.5
   - If your answer is not parsable or incomplete, score -2
- Format Score:
   - If you follow the tag format exactly as above, score 1
   - Otherwise, score -1
You will get the final score as their sum. Example:
(1) The format follows the required structure: +1
(2) The final answer is correct: +2
(3) Total evaluation score: 3
Think carefully, follow the structure, and consider the evaluation rules.`<|im_end|>`

`<|im_start|>`user
{input the puzzle} `<|im_end|>`

`<|im_start|>`assistant
**<think>**

---

**System Prompt and Motivation for Math**

`<|im_start|>`system
You are a helpful assistant. The assistant first thinks about the reasoning process and then provides the user with the answer. Now the user asks you to solve a math problem. After thinking, when you finally reach a conclusion, present the answer in LaTeX format: `\boxed{Your answer}`. i.e., The answer is `\boxed{\frac{14}{3}}`.

You will get evaluated following Evaluation Scoring Rules:
- Correctness Score:
    - If your final answer is correct, score 1
    - If your answer is understandable but wrong, score 0.4
    - If your answer is not parsable or incomplete, score 0
Now, the user will give you the math problem to solve. Think carefully, follow the structure.
`<|im_end|>`

`<|im_start|>`user
{input the question} `<|im_end|>`

`<|im_start|>`assistant
Let's think step by step.

---

**System Prompt and Motivation for CountDown**

`<|im_start|>`system
You are a helpful assistant. The assistant first thinks about the reasoning process in the mind and then provides the user with the answer. The reasoning process and answer are enclosed within <think> </think> and <answer> </answer> tags, respectively, i.e., <think> reasoning process here </think><answer> answer here </answer>. Now the user asks you to solve a math reasoning problem. After thinking, when you finally reach a conclusion, clearly state the equation within <answer> </answer> tags. i.e., <answer> (1 + 2) / 3 </answer>.
You will get evaluated following Evaluation Scoring Rules:
    - if your answer has an equation that uses only the allowed numbers and evaluates exactly to the target, score 1
    - if there's an equation that is readable, but either it uses wrong numbers, or it doesn't evaluate to the target, or it can't be evaluated properly, score 0.4
    - if no equation can be extracted from your answer, score 0
Now, the user will give you the math problem to solve. Think carefully, follow the structure.
`<|im_end|>`

`<|im_start|>`user
{input the question} `<|im_end|>`

`<|im_start|>`assistant

## A.4  CASE STUDY

We provide a case study to illustrate the reasoning capabilities of the final model of **MeRF**. As shown below, the model is capable of providing the correct answer with a clear reasoning process, understanding the motivation of the task and adhering to the rules of the metrics. The model is able to analyze the statements of each character and deduce the truthfulness of their claims, leading to the correct conclusion. The model also provides a clear and concise explanation of the reasoning process, demonstrating its ability to understand and apply logical reasoning in a complex scenario.

---

**A Example of Model Response**

[Puzzle]
A very special island is inhabited only by knights and knaves. Knights always tell the truth, and knaves always lie. You meet 3 inhabitants: Evelyn, Benjamin, and William. Evelyn said that if Benjamin is a knight then William is a knight. In a statement by Benjamin: "Evelyn is not a knave". William commented, "Evelyn is a knight". So who is a knight and who is a knave?
[Model Response]

<think> To solve this, we will analyze each statement based on the rules that knights always tell the truth and knaves always lie.
1. Analyze Evelyn's statement: "If Benjamin is a knight then William is a knight."
   - If Evelyn is a knight, Benjamin must be a knight and William must be a knight. (T->T)
   - If Evelyn is a knave, the statement must be false, so Benjamin being a knight would imply that William is a knave. (F->F)
2. Analyze Benjamin's statement: "Evelyn is not a knave."
   - If Benjamin is a knight, his statement must be true, so Evelyn must be a knight.
   - If Benjamin is a knave, his statement would be false, so Evelyn would be a knave.
3. Analyze William's statement: "Evelyn is a knight."
   - If William is a knight, his statement is true, so Evelyn is a knight.
   - If William is a knave, his statement is false, so Evelyn is a knave.
From Benjamin's statement, if he is a knight, Evelyn must be a knight.
From William's statement, if he is a knight, Evelyn must be a knight.
If Evelyn is a knight, for Benjamin's statement to hold, Benjamin must also be a knight.
This configuration satisfies all statements consistently and rules out other scenarios, leading to all three being knights.
Therefore, Evelyn, Benjamin, and William are all knights. </think>

<answer>
(1) Evelyn is a knight
(2) Benjamin is a knight
(3) William is a knight </answer>

**A Correct Response on Logical Puzzles**

<think>
Let's analyze the statements one by one:
1. Penelope: "Penelope is a knight and Jacob is a knight."
2. Alexander: "Penelope is a knave or Jacob is a knight."
3. Jacob: "Alexander is a knave if and only if Sophia is a knight."
4. Sophia: "Penelope is a knave or Jacob is a knight."

If Penelope is a knight, then both parts of her statement must be true. This would mean she is a knight and Jacob is a knight. However, if Jacob is a knight, and by Sophia's statement, if Jacob is a knight, then Sophia's statement becomes "Penelope is a knave or Jacob is a knight" which is true if Jacob is a knight (regardless of whether Penelope is a knave or knight). This doesn't immediately create a contradiction, so let's try this scenario:
- Assuming Penelope is a knight:
   - For Penelope to be telling the truth, Jacob must be a knight.
   - In this case, Sophia's statement would be true ("Penelope is a knave or Jacob is a knight") since Jacob is a knight.
   - Now, Alexander's statement "Penelope is a knave or Jacob is a knight" would be a true statement because Jacob is a knight.
- However, if Penelope is a knight, Alex's statement is also true, which means he should be a knight but no self-contradiction yet.

Let's now test the scenario with Penelope as a knave:
   - If Penelope is a knave, her statement is false. So, at least one part must be incorrect: either Penelope is not a knight or Jacob is not a knight.
   - If Jacob is a knave, Sophia still has to decide:

1. For Sophia: If Jacob is a knave, Penelope is a knave, then we need to check Alex's and Jacob's statements.
- Alexander: "Penelope is a knave or Jacob is a knight" (false or true). Since it can't be false, we have a true statement.

> - Jacob: "Alexander is a knave if and only if Sophia is a knight" (false if-and-only-if true). Both Alexander and Sophia have to share the same truth identity. Given Penelope as a knave and Jacob as a knave, suggesting Alexander is a knight and Sophia must also be a knight to satisfy the "if-and-only-if" truth statement.
>
> Thus:
> - Penelope: knave
> - Jacob: knave
> - Alexander: knight
> - Sophia: knight
>
> Let's verify:
> - Penelope (knave): "Penelope is a knight and Jacob is a knight" is false.
> - Alexander (knight): "Penelope is a knave or Jacob is a knight" is true.
> - Jacob (knave): "Alexander is a knave if and only if Sophia is a knight" (false if-and-only-if true, meaning Alexander and Sophia are the same, hence true since provided knight).
> - Sophia (knight): "Penelope is a knave or Jacob is a knight" is true.
> All statements hold. </think>
>
> <answer>
> (1) Penelope is a knave
> (2) Alexander is a knight
> (3) Jacob is a knave
> (4) Sophia is a knight </answer>

The above is a correct response from the model after being trained with **MeRF**, with a clear reasoning process and the correct final answer. The highlighted parts in blue show the model's clear understanding of the statements and logical deductions leading to the final conclusion.

The following is an example of a wrong response from the RLVR model in early training stage.

> **A Wrong Response on Logical Puzzles**
>
> <think> We will analyze each statement and compare the implications for each person. From Chloe's statement: "Evelyn is a knight if and only if Riley is a knight". This is a biconditional statement. It means both must be the same (both knights or both knaves). Given this statement, we tentatively conclude directed roles. </think>
> <answer>
> (1) Riley is a knight
> (2) Evelyn is a knight
> (3) Logan is a knight
> (4) Avery is a knight
> (5) Chloe is a knight </answer>

This is a wrong response. The model fails to analyze the puzzle carefully with logical reasoning, only focusing on the first statement and suspected to reward hacking for the format reward, by simply trying to make all the characters knights without verifying the statements.

## A.5 ADDITIONAL RESULTS

Figure 13a is the performance of MeRF and RLVR with reward function of different density (the default by ground-truth reward function is the densest, the medium-sparse is binary + format reward, and the sparse is only binary reward) on K&K Logic Puzzle dataset. For naive RLVR, the performance increases with denser reward function in this case (denser reward function provides more informative reward signal for backward optimization), while MeRF also benefits more from denser reward function (providing more informative motivation signal for forward in-context learning).

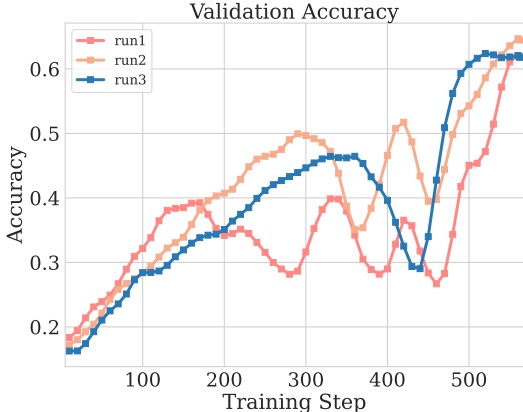

Figure 10: This is multiple runs of MeRF with adverse motivation on K&K Logic Puzzle dataset, showing the similar unstable learning dynamics of the training process, after which the model adapts to discount the motivation signal and achieves better performance.

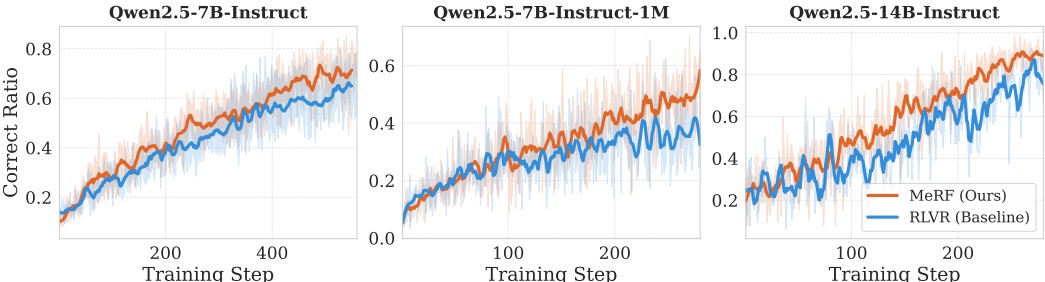

Figure 11: **Correct Ratio** of generated answers of the training set during the training process on K&K Logic Puzzle dataset. MeRF consistently outperforms the RLVR baseline, demonstrating the better exploration ability encouraged by the in-context motivation for the model to get the best reward during the training process.

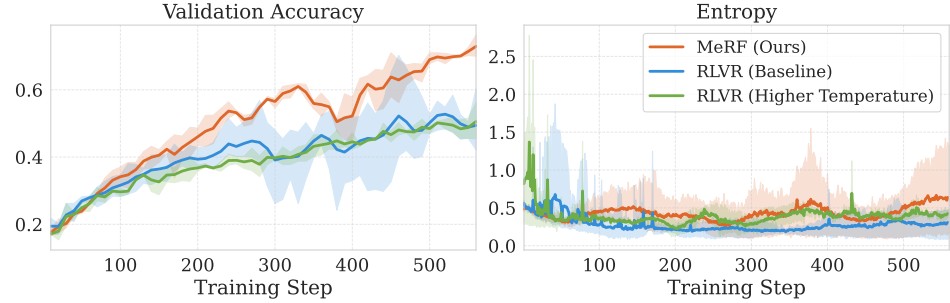

Figure 12: This is the comparison between MeRF, RLVR and RLVR with higher temperature (increase from 1 to 1.2) on K&K Logic Puzzle dataset. Higher temperature leads to higher entropy as expected (comparable with MeRF), but not necessarily a better performance.

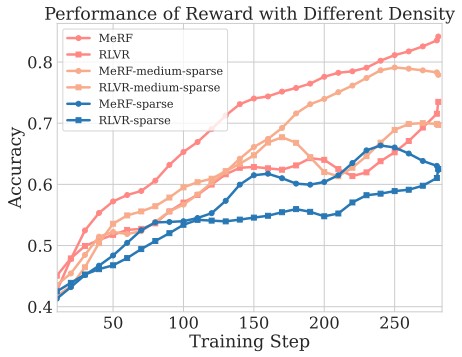
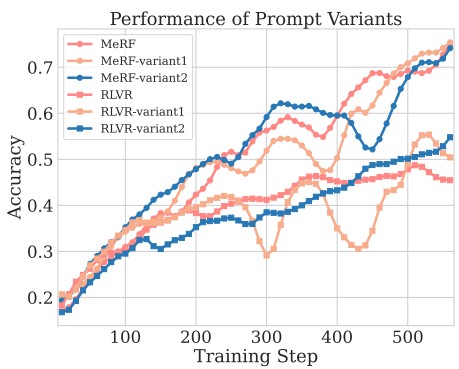

(a) Performance of MeRF and RLVR with reward function of different density.

(b) Performance of MeRF and RLVR with different prompt variants (different presence of motivation).

Figure 13: Performance of MeRF and RLVR with different reward function density and prompt variants.

As shown in the Figure 13b, although the prompt variants lead to some differences on training dynamics, the performances are still significantly connected to the presence of motivation during training, with MeRF (motivation included in the training prompt) consistently outperforming RLVR (no motivation in training prompt) across all prompt variants. This observation further supports the effectiveness of MeRF in leveraging motivations to enhance the RLVR training process, and additionally, highlights the robustness of MeRF to different prompt formulations.

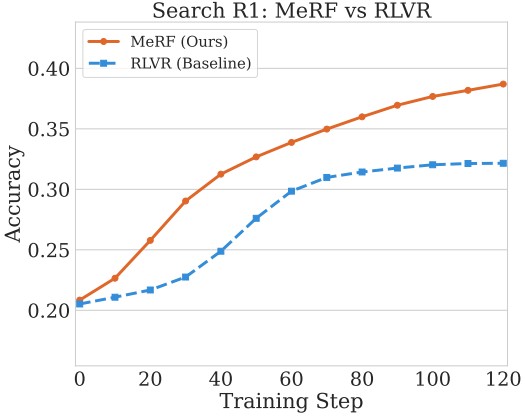

Figure 14: On the agentic task of using search engine for reasoning, MeRF also outperforms RLVR baseline with effective utilization of in-context motivation, demonstrating the generalizability of MeRF to more complex reasoning tasks. Accuracy is evaluated on Natural Questions (NQ)(Kwiatkowski et al., 2019) dataset.

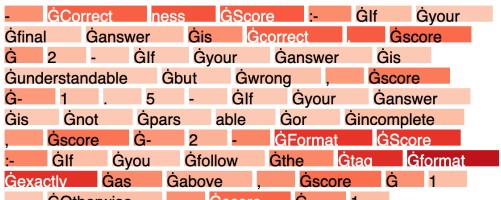

(a) Attention on groundtruth motivation tokens

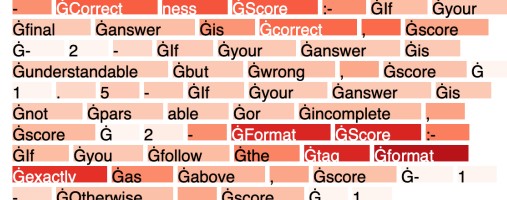

(b) Attention on adverse motivation tokens

Figure 15: This the attention heatmap in the final model of MeRF on K&K Logic Puzzle dataset with different motivations. Attention is calculated when generating <answer>. The model trained with MeRF pays significant attention to the motivation tokens (e.g., "correct", "score", "format", "exactly") when generating the final answer, indicating that the model effectively leverages the motivation information. The comparison between attention weights of groundtruth motivation and adversarial motivation demonstrates that the model learn to distinguish useful information and ignore misleading information in motivations.

