# OpenReview forum: "A Simple "Motivation" Can Enhance Reinforcement Finetuning of Large Reasoning Models"
_ICLR.cc/2026/Conference — ICLR 2026 Poster_

### Official Review · Reviewer_6qk7 · 2025-10-21

**Soundness:** 2
**Presentation:** 3
**Contribution:** 2
**Rating:** 4
**Confidence:** 3

**Summary:**

By providing evaluation criteria with the system prompt (find-grained guideline), RLVR pipeline could be effectively trained.

**Strengths:**

Through a series of experiments, the paper demonstrates that when training a model in a domain where the grading criteria are clearly defined, providing information about the grading scheme alongside the data can significantly accelerate the model’s learning process.

**Weaknesses:**

* As mentioned in the limitations, in situations where there is no prior knowledge of how the grading will be done, an approach that dynamically identifies the motivation and effectively solves the problem seems to have greater scalability.

* Since adding even a suboptimal motivation still provides non-zero additional information, it is unsurprising that the performance improves compared to RLVR. An interesting phenomenon is shown in Fig. 9, where performance increases after 500 steps when an adversarial motivation is provided. It would be important to check whether this result consistently appears across multiple runs, as the paper does not seem to report repeated experiments for this particular setting. Furthermore, when adversarial motivation is given as a guideline, it may be worth analyzing whether the RLVR training process includes any mechanisms that allow the model to ignore or correct such misleading guidance.

**Questions:**

Naturally, ML model could align their answers with the evaluation criteria when grading guidelines are provided rather than only being told whether an answer is correct or not. But does that constitute a discovery?

**Details Of Ethics Concerns:**

I don't have any ethical concern on this paper.

---

> ### Author Response · Authors · 2025-11-21
> **Official Comment by Authors (1/2)**
>
> We sincerely thank the reviewer for the positive and constructive feedback. We appreciate that the reviewer recognized the following strengths of our work:
>
> - The demonstration that in domains with clearly defined grading criteria, providing information about the grading scheme alongside the data can **significantly accelerate** the model's learning process.
>
> The comment is highly encouraging. We have our responses to address all weaknesses and questions below.
>
>
> **[W1] As mentioned in the limitations, in situations where there is no prior knowledge of how the grading will be done, an approach that dynamically identifies the motivation and effectively solves the problem seems to have greater scalability.**
>
> Thanks for the insightful comment. We highly agree with the reviewer that in more general scenarios, dynamically identifying the motivation would enhance the scalability of our approach and introduce more effective RL training for LLMs, and we see it as a promising direction for future work. In this work, we start it with the simpler setting where the reward function is known (RLVR setting) and demonstrate the effectiveness of utilizing dense language motivations(compared to sparse reward signals alone) to improve the current RLVR training paradigm, while the in-context learning ability of LLMs in RLVR has been overlooked at present. We provide extensive experimental evidence and analyses to understand the mechanism and effectiveness of the proposed MeRF method. We believe this work provides a solid foundation for future explorations into more scalable and dynamic motivation identification methods.
>
> Furthermore, we have conducted additional experiments comparing MeRF with tuned-prompt variants using DSPY. Specifically, we optimized the prompt using DSPY and get two more prompt variants for both motivation prompt and base prompt. As shown in the updated *Figure 13b (page 20)*, although the prompt variants lead to some differences on training dynamics, the performances are still significantly connected to the **presence of motivation** during training, with MeRF (motivation included in the training prompt) consistently outperforming RLVR (no motivation in training prompt) across all prompt variants. This observation further supports the effectiveness of MeRF in leveraging motivations to enhance the RLVR training process, and additionally, highlights the robustness of MeRF to different prompt formulations. We believe the way that dynamically tuning prompts during training can further improve our method for more generalizable scenarios in the future work. We appreciate the reviewer for bringing up this insightful point, and we will include these analyses in the paper.
>
> **[W2] An interesting phenomenon is shown in Fig. 9, where performance increases after 500 steps when an adversarial motivation is provided. It would be important to check whether this result consistently appears across multiple runs, as the paper does not seem to report repeated experiments for this particular setting.**
>
> Thanks for the valuable suggestion. We have conducted repeated experiments for the adversarial motivation setting to verify the robustness of the observed phenomenon. As shown in the updated *Figure 10 (Page 19)*, there are very similar unstable phases in the repeated runs, compared to the steadily improving results of ground-truth motivation. This connects the contradiction (between the adversarial motivation and reward signals), with instability in the training process, while the model can eventually overcome and adapt to discount the misleading information in the adversarial motivation and benefit from the true information of the reward structure provided. This phenomenon demonstrates the powerful generalization and in-context learning ability of LLMs, and further highlights the importance and effectiveness of MeRF in leveraging motivations to enhance the RLVR training process.

---

> ### Author Response · Authors · 2025-11-21
> **Official Comment by Authors (2/2)**
>
> **[W3] Furthermore, when adversarial motivation is given as a guideline, it may be worth analyzing whether the RLVR training process includes any mechanisms that allow the model to ignore or correct such misleading guidance.**
>
> Thanks for this insightful comment. We have conducted additional analyses to investigate how the model handles adversarial motivations during training. Specifically, we analyze the attention weights of the model when generating \<answer\> token under adversarial motivation setting. As shown in the updated *Figure 15 (Page 21)*, comparing the attention weights of the model trained with ground-truth motivation and adversarial motivation, we observe that the model try to pay less attention (or ignore) the exactly misleading score and pay more attention to the descriptions about the reward structure (e.g., "correct", "format"). This observation combined with the training dynamics in the adversarial motivation setting, suggests that the model can learn to discount misleading information in the adversarial motivation and focus on more reliable information about the reward structure. We will include these analyses in the paper to provide a more comprehensive understanding.
>
>
> **[Q4] Naturally, ML model could align their answers with the evaluation criteria when grading guidelines are provided rather than only being told whether an answer is correct or not. But does that constitute a discovery?**
>
> Despite its straightforwardness, as we state in the paper, we believe MeRF demonstrates a **significant paradigm shift** in RLVR training (**from implicit/blind exploration then backward updates by sparse rewards, to explicit forward reasoning via dense natural language motivations describing the evaluation rubrics and expected behaviors**), beyond mere intuition. For current RLVR methods, the model learns by randomly exploring the output space and receiving sparse scalar rewards, blindly updating its parameters, overlooking that LLMs possess strong in-context learning capabilities to understand the RL training objectives when provided in natural language.
> The effectiveness of utilizing dense language feedback (motivations) is also supported by prior works on self-evaluation and model-generated text feedback[1,2], which leverage LLMs' in-context learning abilities to improve generation quality through natural language feedback. Our work **showcases the dense language information can be a more direct and effective optimization signal for RLVR than sparse scalar rewards alone.** and we hope it provides the community with a new perspective: **leveraging LLMs' in-context learning ability, incorporating dense language information for better RLVR optimization (e.g., motivations describing reward functions) instead of relying solely on sparse scalar rewards**. We believe this shift is crucial for improving the effectiveness of RLVR methods, as it better aligns with the capabilities of LLMs.
>
> [1] Self-Refine: Iterative Refinement with Self-Feedback
>
> [2] Beyond Sparse Rewards: Enhancing Reinforcement Learning with Language Model Critique in Text Generation

---

> ### Author Response · Authors · 2025-11-21
>
> Thanks again for the constructive comments and suggestions. We have updated the revised manuscript **(all the modifications are marked in blue)**, incorporating all the additional experiments **(starting from Page 19)**. Please kindly refer to the updated version and we look forward to your further feedback.

---

> ### Author Response · Authors · 2025-11-27
> **We kindly look forward to your reevaluation**
>
> Dear reviewer 6qk7,
>
> We are glad that the reviewer appreciates our attempt, and sincerely thank the reviewer for the constructive comments. As suggested, we have additionally conducted repeated experiments for adversarial motivation setting, experiments and analyses on how the model handles adversarial motivations during training, and experiments comparing MeRF with tuned-prompt variants using DSPY. Please let us know if you have other questions or comments.
>
> Since the discussion window has less than a week remaining, we sincerely look forward to your reevaluation of our work and would very appreciate it if you could raise your score to boost our chance of more exposure to the community. Thank you very much!
>
> Best regards,
>
> The authors of MeRF

---

### Official Review · Reviewer_dhpd · 2025-10-25

**Soundness:** 3
**Presentation:** 2
**Contribution:** 3
**Rating:** 6
**Confidence:** 4

**Summary:**

This paper proposes MeRF, a clever approach to improve RLVR for LLMs by injecting a natural language description of the reward function directly into the training prompt, which is termed "in-context motivation". This enables the model to be aware of the optimization objectives during generation, aligning its outputs with desired behaviors more efficiently than the traditional trial-and-error RLVR paradigm, which relies solely on random explorations. Empirical results across benchmarks, including K&K Logic Puzzles, MATH datasets, and CountDown number games, demonstrate substantial gains, and better exploration as evidenced by higher entropy during training. Ablation studies show that performance benefits primarily from the training process rather than inference-time motivation, with MeRF robust to suboptimal or even adverse motivations.

**Strengths:**

- A novel, simple and very practical approach to improve RLVR, which also makes sense
- Interesting experimental design and results on Q4
- Well presented (in terms of design) to make the paper easy to read

**Weaknesses:**

- The experimental results are scattered around the paper and somehow do not seem complete:
  - Figure 1 includes results on 4 different LLMs and Figure 3 includes result on deepseek but most of them not presented in table 1.
  - Results on Figure 2 (right) have no details; What dataset is this?
  - Figure 1, 3, 7, 6, 5, 8 all show increasing performance on steps, but differently grouped (some on metrics, some on datasets), and feels very repetitive, being scattered all around the paper. Need to be better organized
  - Figure 2 (right) and experiment Q3 (figure 8) basically telling the same thing but repeated
  - A number of analysis experiments using different base models and datasets while there is only one single base model for main results, it makes me feel like the models are cherry picked for the analysis experiments.

-  Main results need to be complemented with a number of different base models, to show that the method is robust between LLM choices
- Lots of repetitive figures (in terms of message), system prompt in the main text, a handful of main results, repeated analyses, all these make the paper less dense in terms of how much information it conveys.

- I would expect the improvement of MeRF to highly dependent on what reward function is used in the dataset. What happens if the reward function is much denser (having a lot of different criteria)? What happens if the reward function is less clear in terms of natural language (e.g. Math dataset)? There is no analyses on where the proposed approach would benefit best and where the proposed approach would benefit least.

- system prompts other than K&K puzzle not provided

- Some questionable analyses: continued in questions.

**Questions:**

- Why does MeRF have high-entropy? and is it even a good thing? The paper motivates the need of MeRF like: we need it to have structured exploration instead of naive exploration of usual RLVRs. But entropy is more of a metric for "unstructured exploration"; usually I would expect smaller entropy when we move from unstructured exploration to structured exploration. Why is it not the case here? What happens if we control the temperature to increase entropy: does it help on RLVR?

- It is interesting to see pass@8 saturates fast with RLVR so that it is soon outperformed by MeRF. However, to claim that such early saturation is due to better exploration, I think the authors should also show that the samples generated by RLVR rarely have high rewards (higher than what LLM is currently getting in expectation). Such early saturation might be caused by different training dynamics due to different prompts.

---

> ### Author Response · Authors · 2025-11-21
> **Official Comment by Authors (1/3)**
>
> We sincerely thank the reviewer for the positive and constructive feedback. We appreciate that the reviewer recognized the following strengths of our work:
>
> - A **novel, simple and very practical** approach to improve RLVR, which also makes sense
>
> - **Interesting experimental design** and results on Q4
>
> - **Well presented** (in terms of design) to make the paper **easy to read**
>
> These comments are highly encouraging. We have our responses to address all weaknesses and questions below.
>
>
> **[W1] The experimental results are scattered around the paper and somehow do not seem complete:**
> - **Figure 1 includes results on 4 different LLMs and Figure 3 includes result on deepseek but most of them not presented in table 1.**
> - **Results on Figure 2 (right) have no details; What dataset is this?**
> - **Figure 1, 3, 7, 6, 5, 8 all show increasing performance on steps, but differently grouped (some on metrics, some on datasets), and feels very repetitive, being scattered all around the paper. Need to be better organized.**
> - **Figure 2 (right) and experiment Q3 (figure 8) basically telling the same thing but repeated.**
> - **A number of analysis experiments using different base models and datasets while there is only one single base model for main results, it makes me feel like the models are cherry picked for the analysis experiments.**
>
> Thanks for these very detailed and valuable suggestions.
> - Sorry for the confusion. We didn't include all results in Table 1 due to space limitations in the original manuscript. We have updated *Table 1 (Page 5)* to include the quantitative results for all the models presented in Figure 1 and Figure 3 in the revised version.
> - Figure 2 (right) presents the results on logic puzzles dataset. We will add more details in the figure caption to clarify this.
> - We will reorganize the experimental results section to improve clarity and coherence.
> - Figure 2 (right) and Figure 8 use the same metric (whether validation w/ motivation), but we think they provide different insights: Figure 2 (right) shows the difference between using motivation in training and only in testing, while Figure 8 shows the validation gap does not necessarily lead to a drop in test performance. We will provide clearer explanations to distinguish them.
> - Sorry for the confusion. All the phenomena observed in the analysis experiments are consistent across different base models. The reason we use various base models in the analysis experiments is to demonstrate the robustness of the observed phenomena (rather than focus on a specific model) and we find it is not easy to present them all for each analysis experiment due to space limitations. We will reorganize the presentation of the analysis experiments to make it clearer.
>
> **[W2] Main results need to be complemented with a number of different base models, to show that the method is robust between LLM choices.**
>
> Thanks for this suggestion. We have main results in Table 1 for quantitative comparisons between our method, RLVR baseline, other open-source and commercial models, and more broadly, in Figures (e.g., Figure 1, 3, 4, 5, 6) to show the consistent improvements of MeRF over RLVR across different base models (Qwen2.5 series, DeepSeek-R1-Distill series). We have updated *Table 1 (Page 5)* to include more quantitative comparisons of these models in the revised version.
>
>
> **[W3] Lots of repetitive figures (in terms of message), system prompt in the main text, a handful of main results, repeated analyses, all these make the paper less dense in terms of how much information it conveys.**
>
> Thanks for this comment. Combined with W1 and W2, we will reorganize the experimental results section to improve clarity and coherence.

---

> ### Author Response · Authors · 2025-11-21
> **Official Comment by Authors (2/3)**
>
> **[W4] I would expect the improvement of MeRF to highly dependent on what reward function is used in the dataset. What happens if the reward function is much denser (having a lot of different criteria)? What happens if the reward function is less clear in terms of natural language (e.g. Math dataset)? There is no analyses on where the proposed approach would benefit best and where the proposed approach would benefit least.**
>
> Thanks for the question. In our experiments, logic puzzles have relatively complex reward functions with multiple criteria (reasoning pattern, answer format, correctness), while math tasks have simpler reward functions (mainly correctness). Thus, motivation of logic puzzles can provide more information than math tasks, and we observe that the improvement of MeRF on logic puzzles is larger than that on math tasks. To further analyze the impact of reward function complexity, we have conducted additional experiments on logic puzzles, comparing MeRF with RLVR under different reward function settings, pruning the reward function to be more sparse. Two more reward settings are designed: (1) binary reward (correctness only, most sparse), and (2) binary + format reward (correctness and format, medium sparse). As shown in the updated *Figure 13a (Page  20)*, both RLVR and MeRF benefit from denser reward functions. RLVR performance increases with denser reward function (denser reward function provides more informative reward signal for backward optimization), while MeRF also benefits more from denser reward function (providing more informative motivation signal for forward in-context learning). This observation further supports that the informativeness of the reward function plays a crucial role in the effectiveness of MeRF. We will include these analyses in the paper to provide a more comprehensive understanding of where MeRF benefits most.
>
>
> **[W5] system prompts other than K&K puzzle not provided**
>
> Thanks for pointing this out. We have included the system prompts used in all experiments in the *updated Appendix Pages 15 & 16.*

---

> ### Author Response · Authors · 2025-11-21
> **Official Comment by Authors (3/3)**
>
> **[Q6] Why does MeRF have high-entropy? and is it even a good thing? The paper motivates the need of MeRF like: we need it to have structured exploration instead of naive exploration of usual RLVRs. But entropy is more of a metric for "unstructured exploration"; usually I would expect smaller entropy when we move from unstructured exploration to structured exploration. Why is it not the case here? What happens if we control the temperature to increase entropy: does it help on RLVR?**
>
> Thanks for this insightful comment. We agree that our method encourages structured exploration by the informative motivations, and higher entropy indicates that the model maintains uncertainty to explore more effectively rather than collapsing to suboptimal solutions, which is widely discussed in the community[1,2,3]. But why the structure exploration leads to higher entropy rather than lower entropy? After a careful analysis of the entropy curves in Figures 4 and 6, we find that at the beginning of training, MeRF has lower entropy than RLVR, indicating that the structured exploration introduced by motivations helps the model to focus on more promising areas of the output space compared with naive exploration. As training progresses, RLVR entropy decreases more rapidly, indicating that the model is losing exploration ability and somehow collapsing to suboptimal solutions (e.g., only get format reward) due to sparse reward signal and blind exploration. In contrast, MeRF maintains a relatively higher entropy throughout training, suggesting that the reward space information provided by motivations encourages the model to explore more effectively for the best reward (correct answer, as discussed in your Q7) rather than collapsing to suboptimal solutions rapidly. This observation is also consistent with our analysis on the pass@K improvements of MeRF, which saturates later than RLVR, indicating that MeRF continues to explore and improve over a longer training period.
>
> Temperature control is another way to modify the entropy of the model during generation by adjusting the randomness of token sampling, however, we think it is less effective than guiding structured exploration by motivations. We have conducted additional experiments to compare MeRF with RLVR with higher temperature settings. As shown in the updated *Figure 12 (Page 19)*, increasing temperature of RLVR from 1 to 1.2 does lead to higher entropy, but it does not result in significant performance improvements compared to MeRF, which is also consistent with the analysis in [4].
>
> Thanks for bringing up this insightful question, and we will include these analyses in the paper to provide a more comprehensive understanding of the entropy dynamics in MeRF.
>
>
> [1] The Entropy Mechanism of Reinforcement Learning for Reasoning Language Models
>
> [2] Beyond the 80/20 Rule: High-Entropy Minority Tokens Drive Effective Reinforcement Learning for LLM Reasoning
>
> [3] Reasoning with Exploration: An Entropy Perspective
>
> [4] AceReason-Nemotron 1.1: Advancing Math and Code Reasoning through SFT and RL Synergy
>
>
>
> **[Q7] It is interesting to see pass@8 saturates fast with RLVR so that it is soon outperformed by MeRF. However, to claim that such early saturation is due to better exploration, I think the authors should also show that the samples generated by RLVR rarely have high rewards (higher than what LLM is currently getting in expectation). Such early saturation might be caused by different training dynamics due to different prompts.**
>
> Thanks for this suggestion. To show the exploration improvement brought by MeRF, we think the correct ratio (the ratio of samples achieving the best reward) is a more reasonable metric than the average reward of samples, as the average reward can be influenced by more samples with only format rewards. As shown in the updated *Figure 11 (Page 19)*, MeRF consistently achieves a higher correct ratio than RLVR throughout training, combined with the pass@8 results, demonstrating that MeRF effectively improves structured exploration to find more high-reward samples (correct answers) than RLVR, leading to better performance.

---

> ### Author Response · Authors · 2025-11-21
>
> Thanks again for the constructive comments and suggestions. We have incorporated all the additional experiments and analyses mentioned above into the **revised manuscript (starting from Page 19)**. Please kindly refer to the updated version and we look forward to your further feedback.

---

> > ### Comment · Reviewer_dhpd · 2025-11-25
> >
> > Thank you for resolving most of my concerns and questions. However, my main concern is still about the main text that seems less organized. I will keep my score for now.

---

> ### Author Response · Authors · 2025-11-25
>
> Thanks for the follow-up comment and valuable suggestions on the organization of the main text. We **update the manuscript** and kindly ask the reviewer to check our revised manuscript, where we improve the organization of the main text as detailed below.
> We have reorganized the paper with better presentation and incorporated the suggestions above, as well as some discussion brought up by your and other reviewers' comments.
>
> Specifically,
> - we merge the main curve results into the Figures 1 and the quantitative results into Table 1 to provide a more comprehensive overview of the main results.
> - We update the Table 2 for better presentation.
> - We relocate the tables and figures to group related results together (according to metrics and analysis) and some analysis figures are moved to the appendix to reduce repetition in the main text (e.g., in main results section, Table1 and Table2 together present the quantitative results, Figure1 presents the training dynamics of all the models; in analysis section, each analysis question is answered in a separate subsection with related figures and tables together).
> - We move some redundant and detailed descriptions (e.g., system prompts) to the appendix to improve the flow of the main text.
> - We reformat some figures for better clarity (e.g., Figure 2 and Figure 3).
> - We improve the figure captions and add more explanations in the main text to clarify some confusions (e.g., Figure 2 (right) and Figure 7 (previous 8)).
>
> We believe these changes improve the organization and readability of the paper. We sincerely appreciate the reviewer's valuable feedback, which has helped us enhance the quality of our work and kindly ask the reviewer for further consideration of our revised manuscript (all the modifications are marked in blue).

---

> ### Author Response · Authors · 2025-11-27
> **We kindly look forward to your reevaluation**
>
> Dear reviewer dhpd,
>
> We are glad that the reviewer appreciates our attempt, and sincerely thank the reviewer for the constructive comments. We are happy to hear that most of your concerns and questions have been resolved. As suggested, we reorganized the main text for better presentation as the detailed above. Please let us know if you have other questions or comments.
>
> Since the discussion window has less than a week remaining, we sincerely look forward to your reevaluation of our work and would very appreciate it if you could raise your score to boost our chance of more exposure to the community. Thank you very much!
>
> Best regards,
>
> The authors of MeRF

---

### Official Review · Reviewer_X9Aq · 2025-10-30

**Soundness:** 3
**Presentation:** 4
**Contribution:** 3
**Rating:** 6
**Confidence:** 3

**Summary:**

This paper proposes a simple, low-cost intervention for RL with verifiable rewards (RLVR): append a natural-language description of the reward ("motivation") to each training prompt. The claim is that exposing the policy to the reward structure during rollouts improves exploration (higher entropy, stronger pass@k) and yields consistent gains on K&K logic puzzles, several math benchmarks (AIME/AMC/MATH), and CountDown, even when the motivation text is removed at evaluation. The paper is properly evaluated and ablated.

**Strengths:**

Simple, well-motivated idea that's easy to implement; the paper reads clearly.
Consistent improvements over RLVR across two model families (Qwen2.5, DeepSeek-R1-Distill) and multiple reasoning benchmarks; importantly, performance holds without motivation at test time.
The method achieves better performance in fewer training steps. For example, in one experiment, MeRF achieved better pass@4 and pass@8 performance at step 140 than the final RLVR model did at step 280.

**Weaknesses:**

Currently the MeRF variant is compared only to the RLVR. Given the nature of MeRF consists of injecting the reward in the instruction, consider comparing against tuned-prompt variants via DSPY (https://github.com/stanfordnlp/dspy) , to see whether this benefit comes from better prompting.
The method's effectiveness is tied to tasks where the reward function is verifiable and describable in simple natural language. This limits the scope of MeRF, making it unclear how it would apply to tasks with more complex or non-describable reward signals, such as human preference scores.
The entropy analysis (Figure 4) shows higher entropy for MeRF, interpreted as "better exploration," but higher entropy could also indicate increased uncertainty. Alternative explanations aren't ruled out.

**Questions:**

Can you provide evidence that models actually use the motivation during generation (e.g., attention analysis, probing)?
Catastrophic forgetting / over-alignment: After MeRF training, how does the model perform on unrelated general-purpose tasks? Any drop vs. base/RLVR?
Have the authors analysed how complex the motivation prompt needs to be? How sensitive are results to motivation wording, length, or position? Any robustness sweep?
How much improvement comes from simply having better prompts vs. reward-specific information?
How does performance scale with reward function complexity?

---

> ### Author Response · Authors · 2025-11-21
> **Official Comment by Authors (1/3)**
>
> We sincerely thank the reviewer for the positive and constructive feedback. We appreciate that the reviewer recognized the following strengths of our work:
>
> - The **simple, well-motivated idea** and **clear presentation** of the paper.
>
> - The **consistent improvements** over RLVR across multiple models, and the important finding that performance holds without motivation at test time.
>
> - The **significant efficiency gains**, noting MeRF achieves better performance in fewer training steps.
>
> These comments are highly encouraging. We have our responses to address all weaknesses and questions below.
>
>
> **[W1] Currently the MeRF variant is compared only to the RLVR. Given the nature of MeRF consists of injecting the reward in the instruction, consider comparing against tuned-prompt variants via DSPY (https://github.com/stanfordnlp/dspy), to see whether this benefit comes from better prompting.**
>
> Thanks for the valuable suggestion. We have conducted additional experiments comparing MeRF with tuned-prompt variants using DSPY. Specifically, we optimized the prompt using DSPY and get two more prompt variants for both motivation prompt and base prompt. As shown in the updated *Figure 13b (page 20)*, although the prompt variants lead to some differences on training dynamics, the performances are still significantly connected to the **presence of motivation** during training, with MeRF (motivation included in the training prompt) consistently outperforming RLVR (no motivation in training prompt) across all prompt variants. This observation further supports the effectiveness of MeRF in leveraging motivations to enhance the RLVR training process, and additionally, highlights the robustness of MeRF to different prompt formulations. We believe the way that dynamically tuning prompts during training can further improve our method for more generalizable scenarios in the future work. We appreciate the reviewer for bringing up this insightful point, and we will include these analyses in the paper.
>
>
>
> **[W2] The method's effectiveness is tied to tasks where the reward function is verifiable and describable in simple natural language. This limits the scope of MeRF, making it unclear how it would apply to tasks with more complex or non-describable reward signals, such as human preference scores.**
>
> Thanks for the insightful comment. We highly agree with the reviewer that MeRF is more effective in scenarios where the reward function can be clearly described in natural language (current RLVR settings). As we state in the limitations, we see it as a promising direction for future work to explore more general scenarios where the reward function is complex or non-describable, such as human preference scores, by dynamically identifying motivations. At the same time, we noted that there are some rubric-based papers[1,2,3] that explore using verifiable rubrics as metrics to evaluate human preferences, which makes it possible to describe human preference scores in natural language, and we think it is promising to extend the scope of MeRF in this direction. We believe our current work provides a solid foundation in RLVR settings for future explorations into more scalable and dynamic motivation identification methods.
>
> [1] Rubric-Based Benchmarking and Reinforcement Learning for Advancing LLM Instruction Following
>
> [2] Multi-IF: Benchmarking LLMs on Multi-Turn and Multilingual Instructions Following
>
> [3] Instruction-following evaluation for large language models

---

> ### Author Response · Authors · 2025-11-21
> **Official Comment by Authors (2/3)**
>
> **[W3] The entropy analysis (Figure 4) shows higher entropy for MeRF, interpreted as "better exploration," but higher entropy could also indicate increased uncertainty. Alternative explanations aren't ruled out.**
>
> Thanks for this question. We agree that higher entropy indicates increased uncertainty, and we believe this uncertainty is crucial for better exploration of RL training. The community[1,2,3] has widely discussed the importance of maintaining entropy during RL training to encourage exploration and prevent early entropy collapse, which can lead to suboptimal policies. In our case, the RLVR baseline is blind to reward information during training, more likely to get easier reward (format correct but wrong answer) and fail to get best reward (correct answer) due to sparse rewards, leading to entropy collapse and suboptimal performance (only achieve format correctness). In contrast, MeRF leverages dense language motivations, informing the model about the reward structure and expected behaviors (informing there are more reward the model should try to get). Higher entropy in MeRF indicates that the motivations guide model to more effectively explore for the best reward (correct answer) rather than collapsing to suboptimal solutions, ultimately leading to better performance. Thanks for pointing out and we will add more analyses in the paper to improve the explanation.
>
> [1] The Entropy Mechanism of Reinforcement Learning for Reasoning Language Models
>
> [2] Beyond the 80/20 Rule: High-Entropy Minority Tokens Drive Effective Reinforcement Learning for LLM Reasoning
>
> [3] Reasoning with Exploration: An Entropy Perspective
>
>
> **[Q4] Can you provide evidence that models actually use the motivation during generation (e.g., attention analysis, probing)?**
>
> Thanks for this question. We have conducted additional analyses to investigate how models utilize the motivation during generation. Specifically, we analyze the attention weights of the model when generating \<answer\> token, which represents the beginning of the final answer. As shown in the updated *Figure 15 (Page 21)*, we observe that the model trained with MeRF pays significant attention to the motivation tokens (e.g., "correct", "score", "format", "exactly"), indicating that the model effectively leverages the motivation information when generating the final answer.

---

> ### Author Response · Authors · 2025-11-21
> **Official Comment by Authors (3/3)**
>
> **[Q5] Catastrophic forgetting / over-alignment: After MeRF training, how does the model perform on unrelated general-purpose tasks? Any drop vs. base/RLVR?**
>
> Thanks for this question. We have conducted additional experiments to evaluate the performance of RLVR and MeRF trained models on several general-purpose tasks, including MMLU, hellaswag, and truthfulqa. As shown in the Table below, MeRF and RLVR trained models achieve comparable performance on these general tasks compared to the base models, no significant drop is observed, indicating that MeRF does not lead to catastrophic forgetting or over-alignment to the specific tasks during RL training.
>
> | Models| MMLU (acc)| HellaSwag (acc_norm)| TruthfulQA_gen (bleu_acc)| TruthfulQA_mc1 (acc)| TruthfulQA_mc2 (acc) |
> |-|-|-|-|-|-|
> | Qwen2.5-7B-Instruct| 0.7176 ± 0.0036| 0.8044 ± 0.0040| 0.5043 ± 0.0175| 0.4786 ± 0.0175| 0.6470 ± 0.0155|
> |+RLVR (baseline)| 0.7132 ± 0.0028| 0.8031 ± 0.0035| 0.4987 ± 0.0152| 0.4721 ± 0.0163| 0.6423 ± 0.0148|
> |+MeRF (ours)| 0.7201 ± 0.003| 0.8056 ± 0.0039| 0.5165 ± 0.0175| 0.4737 ± 0.0175| 0.6484 ± 0.0155|
> | Qwen2.5-14B-Instruct| 0.7887 ± 0.0033| 0.8436 ± 0.0036| 0.5190 ± 0.0175| 0.5177 ± 0.0175| 0.6897 ± 0.0149|
> |+RLVR (baseline)| 0.7894 ± 0.0029| 0.8442 ± 0.0034| 0.5231 ± 0.0152| 0.5201 ± 0.0163| 0.6905 ± 0.0148|
> |+MeRF (ours)| 0.7902 ± 0.0042| 0.8451 ± 0.0039| 0.5278 ± 0.0165| 0.5208 ± 0.0155| 0.6912 ± 0.020|
>
>
> **[Q6] Have the authors analysed how complex the motivation prompt needs to be? How sensitive are results to motivation wording, length, or position? Any robustness sweep?**
>
> Thanks for this insightful question. We have conducted additional experiments to analyze the sensitivity of MeRF to different motivation prompt settings. We use several variants of prompts (including motivation and not, as described in A1), which differ in wording, length, and position of the reward information. As shown in the *updated Figure 13b (Page 20)*, MeRF consistently outperforms RLVR across all prompt variants, demonstrating its robustness to different motivation prompt settings. Moreover, we observe that different wording and length of the motivation prompt can slightly affect the performance, but the overall trend remains consistent, indicating that MeRF is not overly sensitive to specific prompt formulations. These observations further support that the in-context learning ability and general understanding of reward structures by LLMs is powerful, playing a crucial role in the effectiveness of MeRF and should draw more attention from the community. We will include these analyses in the paper to provide a more comprehensive understanding of MeRF's robustness.
>
>
>
> **[Q7] How much improvement comes from simply having better prompts vs. reward-specific information? How does performance scale with reward function complexity?**
>
> The question for discussing the contribution of better prompts vs. reward-specific information is answered in Q1 and Q6. For reward function complexity, we conducted additional experiments comparing MeRF with RLVR with different reward function settings, pruning the reward function to be more sparse. Two more reward settings are designed: (1) binary reward (correctness only), and (2) binary + format reward (correctness and format). As shown in the updated *Figure 13a (Page 20)*, For naive RLVR, the performance increases with denser reward function in this case (denser reward function provides more informative reward signal for backward optimization), while MeRF also benefits more from denser reward function (providing more informative motivation signal for forward in-context learning).

---

> ### Author Response · Authors · 2025-11-21
>
> Thanks again for the constructive comments and suggestions. We have updated the revised manuscript **(all the modifications are marked in blue)**, incorporating all the additional experiments **(starting from Page 19)**. Please kindly refer to the updated version and we look forward to your further feedback.

---

> ### Author Response · Authors · 2025-11-27
> **We kindly look forward to your reevaluation**
>
> Dear reviewer X9Aq,
>
> We are glad that the reviewer appreciates our attempt, and sincerely thank the reviewer for the constructive comments. As suggested, we have additionally conducted ablation experiments on prompt variants using DSPY, experiments and analyses on catastrophic forgetting, experiments and sensitivity analyses on motivation prompts, and experiments on reward function complexity. Furthermore, inspired by your comments, we have provided more analyses to clarify the interpretation of entropy results and how models utilize motivations during generation. Please let us know if you have other questions or comments.
>
> Since the discussion window has less than a week remaining, we sincerely look forward to your reevaluation of our work and would very appreciate it if you could raise your score to boost our chance of more exposure to the community. Thank you very much!
>
> Best regards,
>
> The authors of MeRF

---

### Official Review · Reviewer_9PZQ · 2025-11-02

**Soundness:** 1
**Presentation:** 1
**Contribution:** 2
**Rating:** 2
**Confidence:** 5

**Summary:**

The paper introduces Motivation-enhanced Reinforcement Finetuning (MeRF), a method that injects a natural language description of the reward function into the prompt during RLVR training to make LLMs aware of the optimization objective. This leverages in-context learning to improve efficiency over standard RLVR. Contributions include empirical evaluations on logic puzzles and math benchmarks showing performance gains, and analyses on motivation-reward consistency.

**Strengths:**

1. The approach creatively combines in-context learning with RL by explicitly providing reward rules as "motivation," offering a simple extension to existing RLVR paradigms that could inspire hybrid training methods.

2. Experiments cover multiple models (e.g., Qwen2.5 series) , with consistent comparisons to baselines, providing some evidence of improved accuracy and efficiency.

3. The paper is well-structured, with clear illustrations of the method, prompts, and results, making the core idea accessible.

**Weaknesses:**

1. The method is overly simplistic and lacks rigorous theoretical justification; it's unclear how the specific reward scoring rules (e.g., +2 for correctness, -1.5 for understandable but wrong answers) mechanistically influence the model's generation of correct reasoning trajectories, relying too much on intuition without deeper analysis.

2. Extensive experimental data is provided mainly for logic puzzles, but for more general tasks like mathematics and code generation, the motivation descriptions appear ineffective or irrelevant, as evidenced by smaller gains on MATH benchmarks and no code-specific results.

3. Experiments rely heavily on simple numerical comparisons (e.g., accuracy curves), lacking in-depth qualitative analysis, such as case studies of trajectory changes or failure modes, which fails to convincingly support the paper's motivation and leaves readers questioning the underlying mechanisms.

**Questions:**

1. Could the authors provide a theoretical explanation or ablation on how the reward rules in the motivation prompt causally affect trajectory generation? For instance, why do negative scores for "understandable but wrong" answers guide the model better than a binary reward?

2. Why are gains on math tasks (e.g., only 3-4% average improvement) much smaller than on puzzles? Please elaborate on why the method may not generalize to code or other domains, and suggest experiments to test this.


3. The analysis section mentions Q1-Q4 but seems incomplete in the provided document. Can you expand on deeper insights, such as visualizing prompt-motivation interactions or reward hacking examples, to better convince readers of the method's value?

---

> ### Author Response · Authors · 2025-11-21
> **Official Comment by Authors (1/3)**
>
> We sincerely thank the reviewer for the positive and constructive feedback. We appreciate that the reviewer recognized the following strengths of our work:
>
> - The creative combination of in-context learning with RL through "motivation," offering **a simple and inspiring** extension to existing RLVR paradigms.
>
> - The **consistent experimental evidence** of improved accuracy and efficiency across multiple models (e.g., Qwen2.5 series) compared to baselines.
>
> - The **well-structured presentation** of the paper, making the core idea accessible.
>
> These comments are highly encouraging. We have our responses to address all weaknesses and questions below.
>
> **[W1] The method is overly simplistic and lacks rigorous theoretical justification; it's unclear how the specific reward scoring rules (e.g., +2 for correctness, -1.5 for understandable but wrong answers) mechanistically influence the model's generation of correct reasoning trajectories, relying too much on intuition without deeper analysis.
> [Q1] Could the authors provide a theoretical explanation or ablation on how the reward rules in the motivation prompt causally affect trajectory generation? For instance, why do negative scores for "understandable but wrong" answers guide the model better than a binary reward?**
>
> We appreciate the reviewer's insightful question and the acknowledgment of our method's simplicity.
> - [Theoretical justification] Despite its straightforwardness, as we state in the paper, we believe MeRF demonstrates a **significant paradigm shift** in RLVR training (**from implicit/blind exploration then backward updates by sparse rewards, to explicit forward reasoning via dense natural language motivations describing the evaluation rubrics and expected behaviors**), beyond mere intuition. For current RLVR methods, the model learns by randomly exploring the output space and receiving sparse scalar rewards, blindly updating its parameters, overlooking that LLMs possess strong in-context learning capabilities to understand the RL training objectives when provided in natural language.
> The effectiveness of utilizing dense language feedback (motivations) is also supported by prior works on self-evaluation and model-generated text feedback[1,2], which leverage LLMs' in-context learning abilities to improve generation quality through natural language feedback. Our work **showcases the dense language information can be a more direct and effective optimization signal for RLVR than sparse scalar rewards alone.**
>
> - [Analysis on mechanism of MeRF] In the paper section 4, we provide extensive analyses to understand the mechanism of MeRF (and why MeRF works), including entropy, Pass@K improvement, the impact of the validation gap brought by motivation, and comparisons with suboptimal or adversarial motivations. These analyses help us understand how MeRF works and collectively suggest that MeRF effectively improves the RL training process by in-context motivations.
>
> - [Why do negative scores for "understandable but wrong" answers guide the model better than a binary reward?] We have conducted additional ablation experiments comparing MeRF with binary reward settings. The result is shown in the updated *Figure 13a (page 20)*, where RLVR and MeRF with binary reward settings are compared with the original reward settings. We observe that MeRF consistently outperforms RLVR under both reward settings, demonstrating the effectiveness of MeRF. Moreover, we find that the improvement of MeRF over RLVR is larger under the original reward setting than the binary reward setting, indicating that more informative motivations (with negative scores for "understandable but wrong" answers) can further enhance the effectiveness of MeRF. As it is a common way to include this kind of format reward (negative scores for "understandable but wrong") or partial credit in RLVR settings to provide more informative feedback, MeRF can effectively leverage such information with motivation in a more direct way in the context of RLVR training, leading to better performance.
>
> In summary, we hope our work provides the community with a new perspective: **leveraging LLMs' in-context learning ability, incorporating dense language information for better RLVR optimization (e.g., motivations describing reward functions) instead of relying solely on sparse scalar rewards**. We believe this shift is crucial for improving the effectiveness of RLVR methods, as it better aligns with the capabilities of LLMs.
>
> [1] Self-Refine: Iterative Refinement with Self-Feedback
>
> [2] Beyond Sparse Rewards: Enhancing Reinforcement Learning with Language Model Critique in Text Generation

---

> ### Author Response · Authors · 2025-11-21
> **Official Comment by Authors (2/3)**
>
> **[W2] Extensive experimental data is provided mainly for logic puzzles, but for more general tasks like mathematics and code generation, the motivation descriptions appear ineffective or irrelevant, as evidenced by smaller gains on MATH benchmarks and no code-specific results.
> [Q2] Why are gains on math tasks (e.g., only 3-4% average improvement) much smaller than on puzzles? Please elaborate on why the method may not generalize to code or other domains, and suggest experiments to test this.**
>
> - [Why gains on math is less than logic puzzles] Thanks for this question. In the experiments, we observe that MeRF consistently outperforms RLVR baseline across various tasks, including logic puzzles, math benchmarks (AIME24&25, AMC23, and MATH500), and countdown math problems. However, the improvement on math seems not as significant as in logic puzzles, but it is also not trivial (3-4% average improvement over RLVR baseline).
> We believe this is due to two reasons: (1) **The reward function for math tasks is relatively simpler** (sparser and binary, i.e., only correctness matters) than that for logic puzzles (which involves rules about reasoning pattern, answer format, correctness, as a more detailed rubric). Thus, more informative motivations can be provided in logic puzzles than math tasks, leading to larger improvements for the RLVR training. The additional ablation experiment in Q1 also supports this point. Despite less informative motivations, we still observe consistent improvements on math tasks and we believe motivations are not ineffective or irrelevant for math tasks.
> (2) **Math tasks are pre-trained more sufficiently in LLMs** than logic puzzles. As we observe in the Table 1, the base models and even powerful commercial models have quite low accuracy on logic puzzles, showing that logic puzzles are challenging and unseen in the training of most models. In this situation, we believe the results that RLVR effectively improves base models on logic puzzles and MeRF further improves RLVR significantly, more convincingly demonstrating the effectiveness of MeRF.
>
> - [why the method may not generalize to code or other domains] From our point of view, MeRF can be effective in any RLVR setting, where the reward function can be described in natural language (reward function is verifiable and describable). The improvements can be affected by the informativeness of the reward functions. For code generation tasks, there are perfect testing environments (e.g., unit tests) to provide binary correctness rewards, making it suitable for RLVR, but less informative for motivations to describe the reward function. Thus, coding is not the ideal domain to show the effectiveness of our method, and the results are not straightforward to prove our idea.
>
> We conduct an additional experiment on agentic task with search-R1[1] to show whether MeRF is effective in this domain. As shown in the updated *Figure 14 (Page 21)*, MeRF outperforms RLVR baseline consistently, demonstrating the effectiveness of MeRF in the search-engine agent domain, where it shares similar RLVR settings (verifiable reward function described in natural language). We believe MeRF can be effective in other domains with similar RLVR settings, and we encourage future work to explore more general scenarios where the reward function is complex or non-describable, such as human preference scores.
>
> [1] Search-R1: Training LLMs to Reason and Leverage Search Engines with Reinforcement Learning

---

> ### Author Response · Authors · 2025-11-21
> **Official Comment by Authors (3/3)**
>
> **[W3] Experiments rely heavily on simple numerical comparisons (e.g., accuracy curves), lacking in-depth qualitative analysis, such as case studies of trajectory changes or failure modes, which fails to convincingly support the paper's motivation and leaves readers questioning the underlying mechanisms.
> [Q3] The analysis section mentions Q1-Q4 but seems incomplete in the provided document. Can you expand on deeper insights, such as visualizing prompt-motivation interactions or reward hacking examples, to better convince readers of the method's value?**
>
> Thanks for the valuable suggestion. We have added more qualitative analyses to better illustrate the effectiveness of MeRF, including visualizations of attention weights to show how models utilize motivations during generation *(updated Figure 15, Page 21)*, and case studies comparing the correct response and reward hacking examples *(Appendix Page 18)*. In the attention visualization, we observe that the model trained with MeRF pays significant attention to the motivation tokens (e.g., "correct", "score", "format", "exactly") when generating the final answer, indicating that the model effectively leverages the motivation information. The comparison between attention weights of groundtruth motivation and adversarial motivation demonstrates that the model learn to distinguish useful information and ignore misleading information in motivations. In the case studies, correct responses generated by MeRF demonstrate a clear reasoning trajectory with logical steps (e.g., "Assuming...", "Thus, ...", "Let's verify..."). The reward hacking example shows how the response hacks the reward by producing format-correct but totally wrong answers. The analyses provide deeper insights into the mechanism of MeRF and convincingly support the paper's motivation. Thanks for pointing out and we will include these analyses in the paper to improve the explanation.

---

> ### Author Response · Authors · 2025-11-21
>
> Thanks again for the constructive comments and suggestions. We have updated the revised manuscript **(all the modifications are marked in blue)**, incorporating all the additional experiments **(starting from Page 19)**. Please kindly refer to the updated version and we look forward to your further feedback.

---

> ### Author Response · Authors · 2025-11-27
> **We kindly look forward to your reevaluation**
>
> Dear reviewer 9PZQ,
>
> We are glad that the reviewer appreciates our attempt, and sincerely thank the reviewer for the constructive comments. As suggested, we have additionally conducted ablation experiments on reward settings (compare binary reward with original reward) and agentic task experiments (search-R1), and provided more qualitative analyses (attention visualization and case studies) to better illustrate the effectiveness of MeRF. Please let us know if you have other questions or comments.
>
> Since the discussion window has less than a week remaining, we sincerely look forward to your reevaluation of our work and would very appreciate it if you could raise your score to boost our chance of more exposure to the community. Thank you very much!
>
> Best regards,
>
> The authors of MeRF

---

### Author Response · Authors · 2025-12-03
**General Summary to AC**

Dear AC,

Thank you for your dedication to handling a new batch of submissions, and we appreciate it very much. We received valuable feedback from all 4 reviewers, with recognition of our novel, but simple and very practical method to improve RLVR (`[9PZQ, X9Aq, dhpd]`). They also appreciated our extensive and interesting experimental design (`[9PZQ, X9Aq, dhpd, 6qk7]`), as well as the well-presented and easy-to-read paper (`[9PZQ, X9Aq, dhpd, 6qk7]`).

Since the early end of the discussion phase, most reviewers have not had the opportunity to reevaluate their scores after our rebuttal. To facilitate your evaluation, we would like to summarize the main concerns raised by the reviewers and how we have addressed them in the response.

---
**Points raised by reviewers `[X9Aq, dhpd]`:**

Both reviewers provided positive support for our work, acknowledging the novelty and practicality of our method, extensive analysis, and well-presented paper.
- Reviewer `X9Aq` provides very constructive suggestions on **robustness to prompt variants, reward function complexity, and entropy analysis interpretation**. We gained valuable insights from these comments and the additional experiments, helping us improve the quality of our work.
- Reviewer `dhpd` announced **most of their concerns have been resolved** after our responses with a remaining minor concern about the organization of the paper. After that, we updated the revised manuscript, but the discussion window closed before further engagement.
---
**Points raised by reviewers `[9PZQ]`:**

Reviewer `9PZQ` acknowledges our method as a creative, simple, and effective method for RLVR, raising concerns about the **mechanism of how MeRF works with the specified reward scoring rules**. Actually, as one of the core contributions of our work, we provided extensive experimental evidence and analyses in the original manuscript **Section 3 & 4** (e.g., entropy analysis, pass@K analysis) to understand the mechanism and effectiveness of MeRF, demonstrating that MeRF benefits from a more structured exploration process guided by motivations. To further address the reviewer's concerns, we conducted additional experiments comparing MeRF with baseline under different reward function settings and provided more qualitative analyses (e.g., attention visualization and case studies) to better illustrate the effectiveness of MeRF. We also clarified that MeRF demonstrates a **significant paradigm shift** in RLVR training (from implicit/blind exploration then backward updates by sparse rewards, to explicit forward reasoning via **dense natural language motivations**). The idea of language feedback is also supported by prior works on self-evaluation and model-generated text feedback [1,2]. We believe our method is not only intuitive but also strongly supported by extensive experimental evidence and analyses.

The reviewer's opinion that our method is ineffective on math tasks seems to be based on a misunderstanding. In fact, we observed consistent improvements of MeRF over RLVR across all tasks (including math tasks) and the improvements on math tasks is also not trivial (3-4% average improvement over RLVR baseline) compared to the improvements in recent works [3,4].

---
**Point raised by reviewer `[6qk7]`:**

Reviewer `6qk7` raised a concern about the **better scalability of MeRF with dynamic motivation identification**. In our initial manuscript, we discussed this point in the limitations section. We provided further clarification that dynamically identifying the motivation is a promising direction for future work, and in this work, we start with a simpler RLVR setting and provide a solid foundation for future explorations. Moreover, reviewer `6qk7` finds our **experimental results on adversarial motivation interesting** and suggests further analyses. We conducted additional repeated experiments for adversarial motivation setting to verify the robustness of observed phenomenon, and provided more analyses on how the model handles adversarial motivations during training. Ultimately, we further clarified the **significance of the proposed paradigm shift** in RLVR training brought by MeRF in response to the reviewer's question about our discovery.

---
In conclusion, we believe all of the concerns just request further clarification rather than fundamental doubts about the novelty or effectiveness of our method, and we have thoroughly addressed them in our responses and the revised manuscript. We hope this summary is helpful in your evaluation. We sincerely appreciate your time and effort in this process.

Best regards,

Authors

---
[1] Self-Refine: Iterative Refinement with Self-Feedback

[2] Beyond Sparse Rewards: Enhancing Reinforcement Learning with Language Model Critique in Text Generation

[3] The Entropy Mechanism of Reinforcement Learning for Reasoning Language Models

[4] Process Reinforcement Through Implicit Rewards

---

### Meta-Review · Area_Chair_SPf4 · 2026-01-12

**Summary:**

This paper proposes a very simple idea: during RL with verifiable rewards (RLVR), include a natural-language description of the reward function (“motivation” or “rules of the game”) directly in the training prompt. The authors argue that this allows the model to use its in-context learning ability to guide exploration during RL, rather than relying purely on blind trial-and-error.

Across several reasoning benchmarks (e.g. logic puzzles, math) and models, the reviewers generally agree that MeRF consistently improves over an RLVR baseline, often with better sample efficiency, and that the improvement largely remains even when the motivation is removed at test time. During the discussion and rebuttal, the authors added a substantial number of additional experiments and analyses that strengthened the empirical story and addressed many concrete reviewer questions.

Overall, the idea is simple yet effective, though its applicability is limited to settings with clearly describable reward functions, which raises some questions about generalizability.

**Reviewer Concerns:**

The paper validates MeRF primarily on reasoning-focused RLVR tasks where the reward function is verifiable and can be naturally expressed as a textual rubric. Within this setting (logic puzzles, math reasoning, and one agentic reasoning task), the evaluation is reasonably broad and consistent. However, the method’s effectiveness is tied to the availability of informative, describable rewards, and its generality beyond this regime (e.g., preference-based RLHF or domains with opaque rewards) is not demonstrated.

Several reviewers asked whether the gains might simply come from better prompting rather than from exposing the reward structure during training. In response, the authors added experiments with multiple prompt variants and tuned prompts (via DSPy), showing that MeRF consistently outperforms RLVR across these settings. This should mostly address the concern.

One reviewer questions the novelty and mechanism of the approach, noting its simplicity and the lack of a deeper theoretical justification. That said, I believe the simplicity of MeRF is just what makes it appealing and practical.

**Reviewer Scores:**

Likely all unchanged.  Only one reviewer (dhpd, score 6) acknowledged that most concerns were resolved but still expressed reservations about the organization of the main text and mention. The rest of the reviewers didn't engage at all.

---

### Decision · Program_Chairs · 2026-01-26

Accept (Poster)